# Notch2 controls developmental fate choices between germinal center and marginal zone B cells upon immunization

Tea Babushku [1,2], Markus Lechner[1], Stefanie Ehrenberg[1], Ursula Rambold [3], Marc Schmidt-Supprian [2], Andrew J. Yates[4], Sanket Rane[5,7], Ursula Zimber-Strobl [1,6,7] ✉ & Lothar J. Strobl [1,6,7]

Sustained Notch2 signals induce trans-differentiation of Follicular B (FoB) cells into Marginal Zone B (MZB) cells in mice, but the physiology underlying this differentiation pathway is still elusive. Here, we demonstrate that most B cells receive a basal Notch signal, which is intensified in pre-MZB and MZB cells. Ablation or constitutive activation of Notch2 upon T-cell-dependent immunization reveals an interplay between antigen-induced activation and Notch2 signaling, in which FoB cells that turn off Notch2 signaling enter germinal centers (GC), while high Notch2 signaling leads to generation of MZB cells or to initiation of plasmablast differentiation. Notch2 signaling is dispensable for GC dynamics but appears to be re-induced in some centrocytes to govern expansion of IgG1[+] GCB cells. Mathematical modelling suggests that antigen-activated FoB cells make a Notch2 dependent binary fate-decision to differentiate into either GCB or MZB cells. This bifurcation might serve as a mechanism to archive antigen-specific clones into functionally and spatially diverse B cell states to generate robust antibody and memory responses.

Mature splenic B2 cells are comprised of two distinct cell populations: Follicular B (FoB) and Marginal Zone B (MZB) cells. In mouse, FoB cells recirculate between lymphoid organs, whereas MZB cells are mostly sessile in the spleen, where they migrate extensively between the MZ and the follicle, with approximately 40% of MZB cells exchanging between the MZ and follicle within one hour[1]. MZB cells respond to T-cell-independent (TI)-antigens by differentiation into low affinity IgM-producing plasma cells (PC)[2,3]. In contrast, FoB cells respond to T-cell-dependent (TD)-antigens, giving rise to germinal centers (GC), where they undergo class switching and somatic hypermutation, resulting in the generation of high-affinity Ig-switched PCs and memory B cells[4,5].

B cell development begins in the bone marrow (BM) and is completed in the spleen. Immature B cells leave the BM upon successful rearrangement of the B cell-receptor (BCR) and pass through short-lived developmental stages, the transitional B cells (T1–T3), before differentiating into either FoB or MZB cells in the spleen[6–8]. MZB cells may also develop from FoB cells, in particular FoB-II cells, a recirculating long-lived population with an IgM[high]IgD[high]CD21[int]CD23[+] phenotype[9–13]. We recently strengthened this hypothesis by showing

[1]Research Unit Gene Vectors, Research Group B Cell Development and Activation, Helmholtz Zentrum München, German Research Center for Environmental Health, Feodor-Lynen-Str. 21, D-81377 Munich, Germany. [2]TranslaTUM, Center for Translational Cancer Research, Technical University of Munich, Einsteinstraße 25, D-81675 Munich, Germany. [3]Institute of Asthma and Allergy Prevention, Helmholtz Zentrum München, German Research Center for Environmental Health, Feodor-Lynen-Str. 21, D-81377 Munich, Germany. [4]Department of Pathology and Cell Biology, Columbia University Irving Medical Center, 630 West 168th Street, New York, NY 10032, USA. [5]Irving Institute for Cancer Dynamics, Columbia University, 1190 Amsterdam Ave, New York 10027, USA. [6]Institute of Lung Health and Immunity (LHI), Helmholtz Munich, Comprehensive Pneumology Center (CPC-M), Member of the German Center for Lung Research (DZL), Ingolstädter Landstraße 1, 85764 Neuherberg, Germany. [7]These authors contributed equally: Sanket Rane, Ursula Zimber-Strobl, Lothar J. Strobl. ✉e-mail: strobl@helmholtz-muenchen.de

that the adoptive transfer of highly purified FoB cells into immuno-competent recipients led to the generation of donor-derived MZB cells via an intermediate MZ precursor stage[14]. Our data provided strong evidence that FoB cells can act as precursors for MZB cells and we postulated that this could be physiologically mediated via activation of Notch2.

Notch2 signaling has been long known to be a crucial player in MZB cell development. Constitutively active Notch2 signaling in T1 or FoB cells strongly induces MZB cell differentiation[14,15]. Correspondingly, homozygous or heterozygous inactivation of Notch2 in B lymphocytes or of its ligand Delta Like Canonical Notch Ligand 1 (Dll-1) in non-hematopoietic cells leads to the total loss or reduction of MZB cells, respectively[16,17], suggesting a dependence of MZB cell development on ligand/receptor dosage. This phenomenon may imply that the bottleneck in MZB cell development is the extent of Notch2 signaling, which is modulated by Notch2 surface expression and its modification[17–19], as well as the availability of Dll-1-expressing fibroblasts[20]. Additionally, BCR-signaling strength was suggested to influence MZB cell fate decisions[6,21]. We have shown that Notch2 surface expression is strongly upregulated by BCR-stimulation in vitro[14] and others found that BCR-signaling induces recruitment of Adam10 to the plasma membrane, enhancing the cleavage and subsequent activation of Notch2[22]. These findings suggest that BCR signals act upstream of Notch2 signaling, thereby increasing the likelihood of B cells to acquire a strong Notch2 signal that drives their commitment to the MZB cell fate. Interestingly, BCR and Notch2 signaling all converge on Akt signaling, which recently has been shown to drive MZB cell development[23].

While we recently demonstrated that prolonged Notch2 signaling can trigger the trans-differentiation of FoB cells into MZB cells, the physiological regulation of this trans-differentiation process remained unclear.

In this study, we demonstrate that MZB cells are routinely derived from FoB cells during TD-immune responses. Our findings suggest that the strength of Notch signaling plays a crucial role in determining the fate of antigen-activated B cells. B cells receiving strong Notch2 signals develop into either MZB cells or PCs, while antigen-activated B cells which downregulate the Notch2 signal differentiate into GCB cells, which subsequently give rise to long-lived PCs and memory B cells. Therefore, the Notch2 signaling pathway emerges as a pivotal switch in generating functionally and spatially diverse B cell clones with distinct roles in early, late, and recall responses.

## Results

### All B cells receive a Notch signal under steady-state conditions

To identify the B cell populations that receive a Notch signal, we analyzed B lymphocytes in the BM and spleen of CBF:H2B-Venus mice, which express Venus fused to histone 2B (H2B) in all cells that undergo active or have recently experienced Notch signaling[24]. We found that almost all B cells express H2B-Venus. H2B-Venus expression gradually increased during B cell maturation and reached highest expression in MZB cells (Fig. 1a, b, Supplementary Fig. 1a, b). Analysis of the expression of three classical Notch-target genes *Deltex1 (Dtx1)*, *Hes1*, and *Complement receptor 2 (Cr2)* revealed that they are less expressed in FoB cells from Notch2-deficient mice compared with controls (Fig. 1c) and more highly expressed in MZB cells compared with FoB cells, (Supplementary Fig. 1c), confirming that FoB cells receive a basal Notch2 signal that is amplified in MZB cells. The upregulation of H2B-Venus and Hes1 started in pre-MZB cells (CD21^high CD23^+) (Fig. 1d), corroborating our assumption that enhanced Notch2 signaling in T2 or FoB cells initiates MZB cell differentiation.

Although some mature MZB cells exhibited the highest H2B-Venus expression within all B cell subpopulations, approximately half of them expressed less H2B-Venus compared to the majority of B cells in the spleen (Fig. 1d). In contrast, the expression of the Notch-target

gene Hes1 remained stable in all MZB cells (Fig. 1d). Hes1 levels were even faintly increased in H2B-Venus^low MZB cells in comparison to H2B-Venus^high MZB cells and were significantly higher than in FoB cells (Fig. 2a). Both H2B-Venus^high and H2B-Venus^low cells were found within the follicle as well as in the MZ (Fig. 2b). In the follicle, some of the H2B-Venus^high cells had a MZB cell shape (bigger in size and less roundish[1], white arrows). To further characterize H2B-Venus^high and H2B-Venus^low MZB cells we subdivided pre-MZB and MZB cells according to their CD23^+ levels, which are sequentially downregulated during MZB cell maturation[14]. Both the H2B-Venus intensity in the H2B-Venus^high cell fraction and the percentage of H2B-Venus^low cells increased sequentially from pre-MZB, CD23^+/low, and CD23^low MZB cells (Fig. 2c). Additional stainings indicate that H2B-Venus^low MZB cells exhibit a more mature MZB cell phenotype (lower CD23 and IgD levels and higher CD1d and IgM levels) and a marginally higher percentage of apoptotic cells compared with H2B-Venus^high MZB cells (Supplementary Fig. 2a–c)[11,14]. Based on these data, we conclude that the amount of H2B-Venus increases during MZB cell development, which underlines the importance of Notch2 signaling in MZB cell development. However, a subset of these mature MZB cells appears to undergo a reduction or loss of H2B-Venus during their further maturation or when on the path towards cell death/apoptosis. In contrast to FoB cells, MZB cells exhibit self-renewal capacity and an intrinsic capability for plasma cell differentiation. To understand why H2B-Venus levels are lower in certain MZB cells but remain unchanged in FoB cells, we investigated whether this loss occurs through cell division and differentiation processes. Ex vivo isolated splenic B cells were labeled with the proliferation dye CellTrace Far Red and stimulated with LPS, inducing both proliferation and PC differentiation. Plotting CellTrace Far Red against H2B-Venus revealed that the majority of cells lose H2B-Venus during proliferation, with a small subset maintaining high H2B-Venus expression despite proliferation (Supplementary Fig. 3a). We assume that these cells contain Notch-IC in the nucleus and produce new H2B-Venus upon proliferation. Next, we analyzed if the loss of H2B-Venus occurred not only through proliferation but also during PC differentiation. Gating cells based on the same CellTrace Far Red intensity and dividing them into different levels of CD138, which is progressively upregulated during PC differentiation, revealed a correlation between increasing CD138 levels and higher percentages of H2B-Venus^low cells in cells which have the same division rate (Supplementary Fig. 3b). These findings imply a further loss of H2B-Venus expression during PC differentiation.

Based on these findings, we propose that MZB cells receive a strong Notch2 signal during their generation. However, the inherent ability of MZB cells to proliferate and differentiate into PCs results in a gradual loss of H2B-Venus in these cells. Supporting this, H2B-Venus^low MZB cells exhibit a higher percentage of Ki-67^+ cells and elevated levels of Interferon regulatory factor 4 (Irf4) and B-lymphocyte-induced maturation protein-1 (Blimp1), compared to FoB cells and H2B-Venus^high MZB cells, indicating more robust proliferation and stronger differentiation toward plasma cells (Supplementary Fig 3c, d). These results indicate that in MZB cells H2B-Venus is progressively lost due to the combined effects of proliferation and PC differentiation.

### Notch2 signaling is dispensable for GCB cell dynamics but appears to regulate the expansion of IgG1^+ cells in the GC

The H2B-Venus signal was progressively downregulated in germinal center B (GCB) cells (Fig. 3a, Supplementary Fig. 4a). In immunohistochemistry, germinal center structures were predominantly H2B-Venus^− (Fig. 3b), suggesting that Notch signaling is switched off in GCB cells. This observation is consistent with a previous report by Valls and colleagues, who showed that Notch2 and Notch2 pathway genes are transcriptionally silenced by Bcl6 in GCB cells[25]. By day 14, H2B-Venus^high GCB cells were enriched in the light zone (LZ), while H2B-Venus^mid and H2B-Venus^neg cells were predominantly found in the

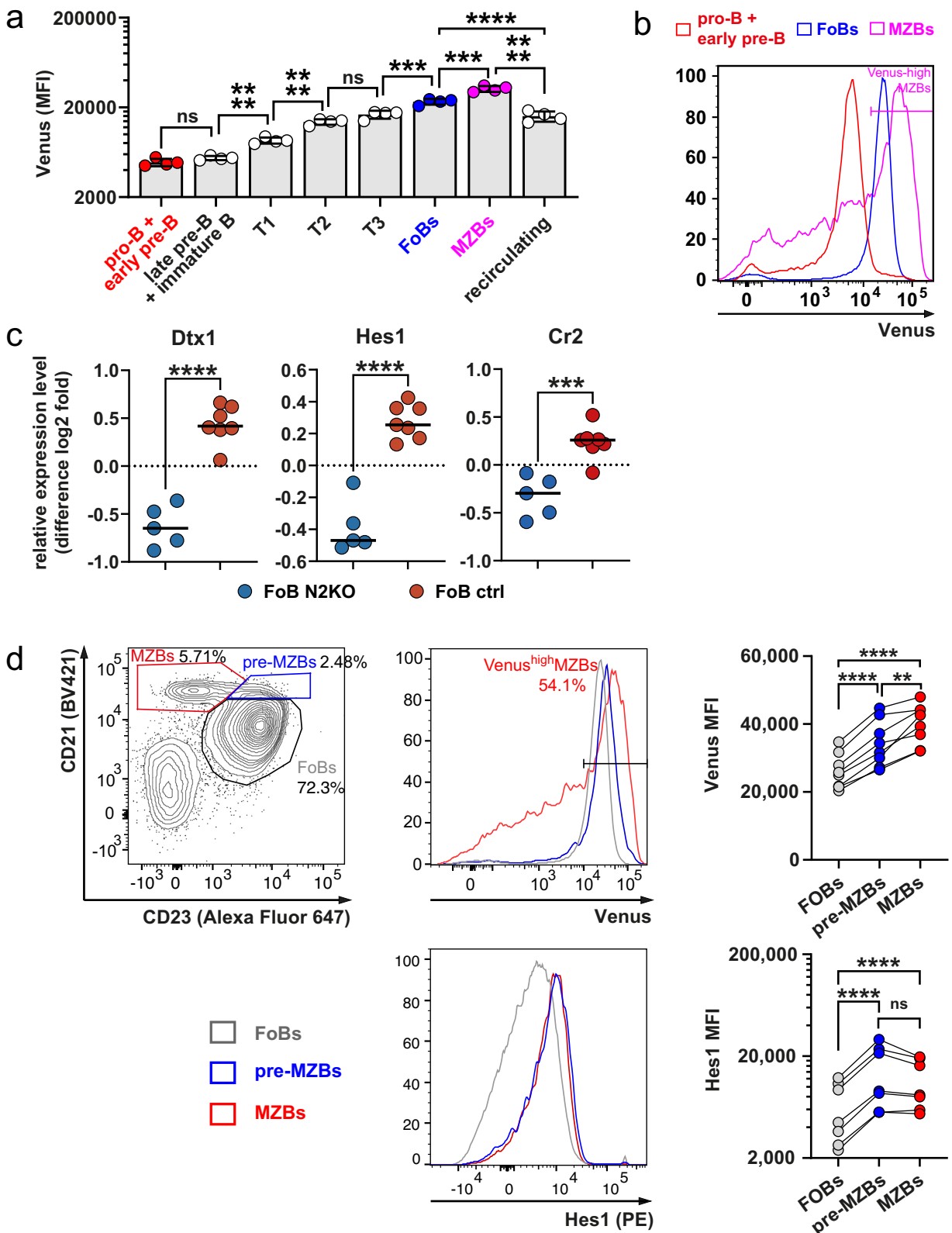

dark zone (DZ) (Supplementary Fig. 4b). H2B-Venus expression and Notch2 expression were slightly higher in LZ-GCB than in DZ-GCB cells (Supplementary Fig. 4c, d). These data suggest the down-regulation of Notch2 signaling in GCB cells, but reactivation in some LZ-B cells.

To validate this finding, we immunized transgenic mice, in which Notch2 is inactivated (N2KO//CAR) or constitutively activated (N2IC/hCD2) upon TD-immunization by employing the Cγ1-Cre strain[26]. As controls, we used control/CAR mice, in which the human coxsackie/adenovirus receptor (CAR) is expressed as reporter upon Cre-mediated recombination (Cre-reporter gene)[27]. The Cre-reporter gene CAR was used to trace B cells that successfully underwent Cre-mediated recombination in N2KO//CAR and control/CAR mice, while hCD2 reported Notch2IC expression in N2IC/hCD2 mice

**Fig. 1 | All B cells are recipients of a Notch signal, which gradually increases during B cell development and is highest in Marginal Zone B cells. a** The graph compiles the Median Fluorescence Intensity (MFI) of H2B-Venus in different B cell populations from CBF:H2B-Venus mice in the bone marrow (BM) and the spleen. The B cell populations were gated as shown in Supplementary Fig. 1a, b. For Marginal Zone B (MZB) cells the MFI for the H2B-Venus[high] peak, as gated in (**b**) is depicted. The bar graph shows mean values, standard deviations (SD), and individual data points from $n = 4$ mice (1 male (m), 3 females (f)). Data were logarithmized before analysis (***$p = 0.0005$, ****$p < 0.0001$), ordinary one-way ANOVA, Tukey´s multiple comparison test. **b** The histogram overlay shows the H2B-Venus expression in the indicated gated cell populations (red: pro/early preB; blue: Follicular B (FoB); pink: MZB). **c** The graphs compile the relative expression levels of the known Notch-target genes *Dtx1* (*Deltex1*), *Hes1*, and *Cr2* (*Complement receptor 2* = CD21) in FoB cells from N2[fl/fl]//CD19-Cre (N2KO) (blue) $n = 5$ mice, and controls (ctrl: CD19-Cre) (red) $n = 7$ mice. The scatter plots show individual data points with means. Unpaired two-tailed *t*-tests were performed (***$p = 0.0005$, ****$p < 0.0001$). The data are extracted from an Illumina Microarray-based gene expression profiling described in the thesis of S. Ehrenberg (available from https://doi.org/10.5282/edoc.19322)[64] and are available at [61]. **d** MFIs of H2B-Venus and Hes1 in FoB (gray), pre-MZB (blue), and MZB (red) cells, which were gated as shown in the contour plot (left). Representative histogram overlays for H2B-Venus are shown in the middle. MFIs for H2B-Venus in MZB cells were only determined in the gated population of the histogram (H2B-Venus[high]). After log transformation data were analyzed by a repeated measures (RM) one-way ANOVA, Tukey´s multiple comparison test $n = 8$ mice (4 m, 4 f) for H2B-Venus: ****$p < 0.0001$, **$p = 0.0067$; $n = 7$ mice (5 m, 2 f) for Hes1: ****$p < 0.0001$. Source data from (**a**), (**c**), and (**d**) are provided as a Source Data file.

(Supplementary Fig. 5a). Deletion efficiency of the targeted regions was comparable among the three mouse strains and percentages of Cre-reporter[+] cells peaked at day 14 at approximately 5-6% (Supplementary Fig. 5b). While all mice had similar percentages of Cre-reporter[+] B cells, N2IC/hCD2 mice failed to generate GCB cells (Fig. 3c). The few GCB cells observed in N2IC/hCD2 mice were mostly Cre-reporter[–] (Fig. 3d) and formed small GC structures not located at the T/B cell border (Fig. 3e). These findings align with previous data from us and Valls and colleagues, showing that constitutive Notch2IC expression is incompatible with the GC reaction[15,25].

In contrast, N2KO//CAR mice generated a similar percentage of GCB cells as control/CAR mice and formed normal GC structures correctly localized at the B/T cell boundary (Fig. 3c–e), demonstrating that Notch2 signaling is not required for GC dynamics. Considering that H2B-Venus expression increased again in some LZ-B cells (Supplementary Fig. 4c), we wondered whether Notch signaling has a function there. Indeed, at day 14 after immunization, the proportion of Cre-reporter[+]IgG1[+] B cells with a GCB phenotype was significantly reduced in N2KO//CAR in comparison to control mice (Fig. 3f, g), suggesting that lack of Notch2 signaling impairs the expansion and/or positive selection of IgG1-switched cells in the LZ.

### Notch2 surface expression and signaling is upregulated in activated B cells

To better understand the regulation of Notch signaling during the TD-immune response, we investigated whether Notch signaling is enhanced in activated B cells. Combined CD40- and BCR-stimulation in vitro led to an around 20-fold upregulation of Notch2 surface levels. This upregulation started early and peaked between 24 h and 48 h (Fig. 4a, Supplementary Fig. 6a). We next examined whether during a TD-immune response Notch2 expression and Notch signaling are increased in activated B cells but are downregulated in GCB cells again. Indeed, Cre-reporter[+]CD38[+]CD95[high] cells, which contain both activated FoB and memory B cells (Supplementary Fig. 6b), displayed the highest Notch2 expression, while GCB cells exhibited the lowest Notch2 levels (Fig. 4b, Supplementary Fig. 6c), which further decreased over time post-immunization (Fig. 4b). Similarly, Hes1 expression was upregulated in Cre-reporter[+]CD95[high]CD38[+] B cells from control/CAR, but not N2KO//CAR mice. Hes1 expression in N2IC/hCD2+ B cells exhibited similar regulation as in control B cells, but was generally higher (Fig. 4c, Supplementary Fig. 6d). Thus, our in vitro and in vivo data suggest that Notch2 surface expression and -signaling are induced in activated FoB cells.

To gain further insights into the effects of sustained Notch2 signaling in activated B cells, we examined the phenotype of Notch2IC-expressing B cells in more detail. N2IC/hCD2[+] B cells were CD38[high] and CD95[high-mid] (Fig. 4d, e). Amounts of Irf4 and Bcl6, the key regulators of PC- and GCB cell differentiation, respectively[28,29], revealed an inverse regulation in N2IC/hCD2 and control/CAR mice. Control/CAR[+] B cells strongly upregulated Bcl6 expression while

maintaining low levels of Irf4, consistent with a GCB cell phenotype. In contrast, N2IC/hCD2[+] B cells failed to upregulate Bcl6 but showed an induction of Irf4 (Fig. 4f). These findings indicate that sustained Notch2 signaling interferes with GC formation by suppressing Bcl6 and inducing Irf4.

Among the N2IC/hCD2[+] B cells a subset exhibited high Irf4 levels, which were also observed in control/CAR mice, but at a significantly lower frequency. The Irf4[high] cells were uniformly B220[–] in both genotypes (Fig. 4g). Histology revealed that the enriched Irf4[high] cells in N2IC/hCD2 mice were predominantly located in the extrafollicular regions, where plasmablastic cells (PB) and short-lived PCs are typically found (Fig. 4h). N2IC/hCD2 mice exhibited increased numbers of 4-hydroxy-3-nitrophenylacetyl (NP)-specific IgM antibody-secreting cells (ASC) in the spleen and elevated IgM titers in the serum 14d after immunization. In contrast, NP-specific IgG1 and IgG3 ASCs, as well as serum titers were decreased (Supplementary Fig. 7a). These findings suggested that Notch2 signaling enhances the production of short-lived IgM PCs.

### Notch2 signaling in activated FoB cells initiates plasma cell differentiation but has to be switched off to enable their terminal differentiation

To confirm a potential role of Notch2 signaling in promoting PC differentiation, we analyzed N2KO//CAR//Cγ1-Cre and N2KO//CD19-Cre mice in comparison to control mice at different time points after immunization. Seven days after immunization, the frequencies of splenic PC were slightly reduced in N2KO compared to controls, while there was no difference at later time points (Supplementary Fig. 7b, c). ELISpot-analyses also revealed a slight reduction of NP-specific ASCs (mainly NP-IgG1) at day 7 (Supplementary Fig. 7d, e), suggesting that Notch2 signaling positively influences the early wave of PC generation, in which mainly short-lived PCs are generated.

To further understand the impact of Notch2 signaling on PC differentiation, we examined Blimp1 levels in Irf4[high]B220[low] cells, which increase during PC differentiation. In N2IC/hCD2 mice, the fraction of Irf4[high]B220[–]Blimp1[high] cells was significantly lower than in controls, and Ki-67 levels were higher (Fig. 5a, b). Although this regulation was not apparent in N2KO//CAR mice (Fig. 5b, Supplementary Fig. 8a), which might be due to a redundant function of Notch1- and Notch2 signaling in PC differentiation, with the major contribution of Notch1, the data suggest that persistent Notch signaling inhibits PC terminal differentiation, keeping PC precursors in a proliferative stage. This led us to hypothesize that downregulation of Notch signaling is necessary for final PC differentiation. To experimentally confirm this, we analyzed the H2B-Venus expression in PCs derived from CBF:H2B-Venus mice. Most TACI[+]CD138[+] PCs lost their H2B-Venus expression (Fig. 5c), with remaining H2B-Venus[high] cells showing partially a (pre-) plasmablast phenotype (CD19[+]B220[+]), while H2B-Venus[mid+low] PCs exhibited full maturation into PCs (CD19[–]B220[–]) (Fig. 5c, d). We concluded that Notch signaling is active in activated B

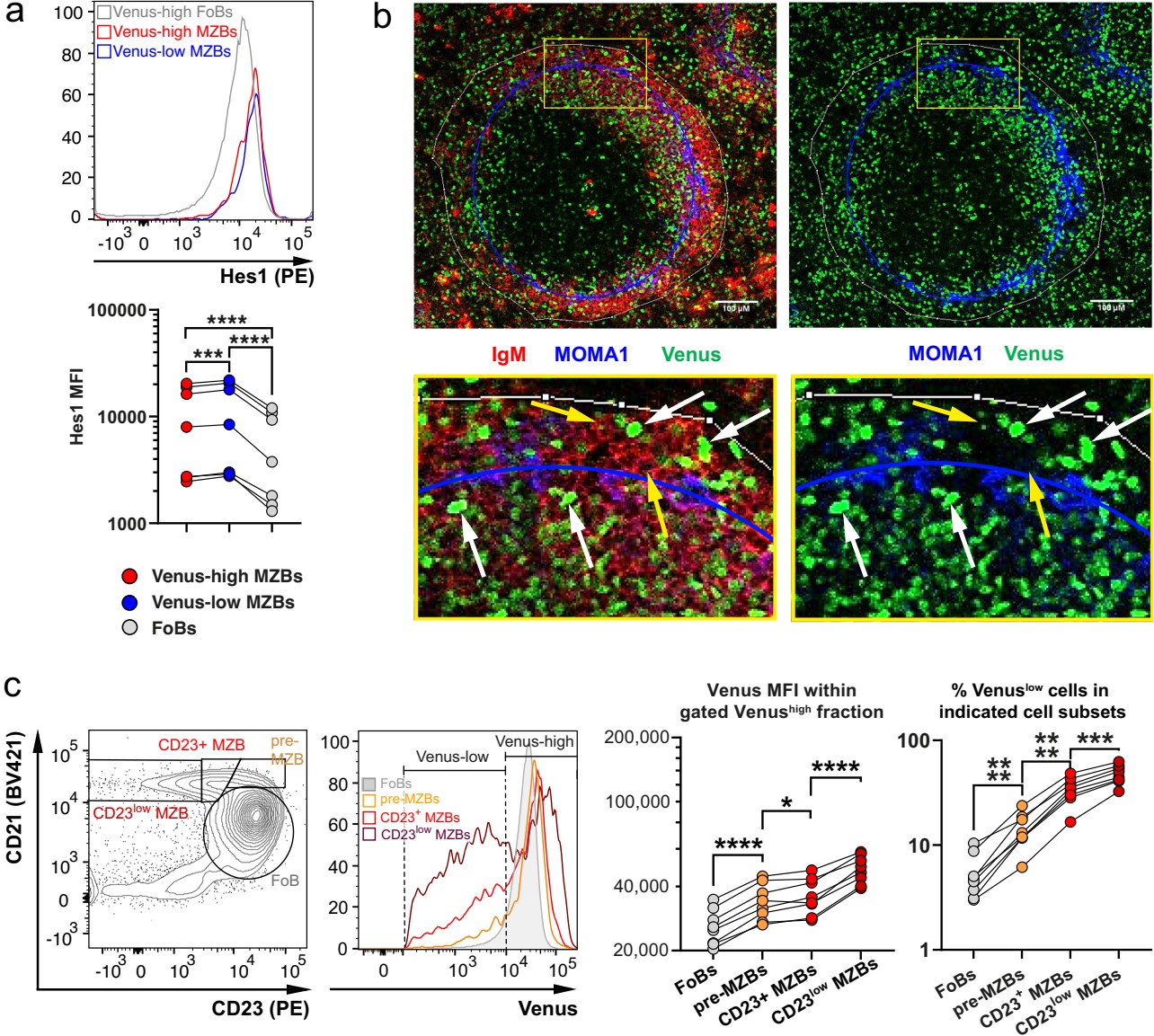

**Fig. 2 | Some Marginal Zone B cells lose their H2B-Venus signal. a** Expression of Hes1 was determined in H2B-Venus$^{low}$ (blue) and H2B-Venus$^{high}$ (red) Marginal Zone B (MZB) cells, as well as in Follicular B (FoB) (gray) cells. Gating of H2B-Venus$^{low}$ and H2B-Venus$^{high}$ cells within the MZB cell gate is shown in Supplementary Fig. 2a. A representative histogram overlay is shown. The graph summarizes the Median Fluorescence Intensity (MFI) of Hes1 in the indicated populations from $n = 7$ mice (5 males (m), 2 females (f)). After log transformation of data, a repeated measures (RM) one-way ANOVA with Tukey´s multiple comparison test was performed (****$p < 0.0001$; ***$p = 0.0003$). **b** Immunofluorescence analysis of splenic sections stained for IgM$^+$ (red) and metallophilic macrophages Moma1$^+$ (blue). Venus signals appear in green. The scale bar represents 100 μM. Images are representative for $n = 3$ CBF:H2B-Venus mice. The faint white lines mark the outer border of the marginal zone (MZ), the blue lines indicate the marginal sinus. Yellow arrows mark

H2B-Venus$^{−/low}$ cells and white arrows H2B-Venus$^{high}$ cells with a MZB cell shape. The yellow squares indicate the area which was magnified. **c** MZB cells were divided into pre-MZB cells (CD21$^{high}$CD23$^{high}$) (orange), newly generated MZB cells (CD21$^{high}$CD23$^+$) (bright red), and more mature MZB cells (CD21$^{high}$CD23$^{low}$) (dark red), as well as FoB cells (CD21$^{mid/low}$CD23$^+$) (filled gray). A representative overlay of the H2B-Venus expression in these gated populations is shown. The graphs illustrate the H2B-Venus$^{high}$ MFIs in the H2B-Venus$^{high}$ fractions and the percentages of H2B-Venus$^{low}$ cells in the indicated populations, $n = 8$ mice (4 m, 4 f). After log transformation of data, RM one-way ANOVAs with Sidak´s multiple comparison tests were performed (H2B-Venus$^{high}$ MFIs: ****$p < 0.0001$, *$p = 0.0393$; H2B-Venus$^{low}$ percentages: ****$p < 0.0001$, ***$p = 0.0003$). Source data from (**a**) and (**c**) are provided as a Source Data file.

cells that are committed to PC differentiation and is completely shut down in mature PCs. To support this conclusion, an in-silico analysis of Notch2 and Notch-target gene expression was performed using the immgen.org database, which showed a significant reduction in Notch2, Hes1, Hes5, and Dtx1 mRNA in PB and PCs compared with FoB and MZB cells (Supplementary Fig. 8b). Collectively, these findings indicate that Notch signaling maintains pre-plasmablasts in a proliferative state, promoting their expansion and must be shut-off to enable terminal PC differentiation.

## Some antigen-activated B cells differentiate into MZB cells during the TD-immune response

The B220$^+$Irf4$^{mid}$ population contained mainly GCB cells (Bcl6$^+$) in control/CAR and non-GCB cells (Bcl6$^-$) in N2IC/hCD2 mice (Fig. 6a, b). Given the previously established role of constitutively active Notch2 signaling in the differentiation of FoB into MZB cells under steady-state conditions[14], we analyzed whether the Cre-reporter$^+$B220$^+$Irf4$^{mid}$ non-GCB cells from N2IC/hCD2 mice acquire a MZB cell phenotype over time. Indeed, by day 7 post-immunization, approximately one-fourth

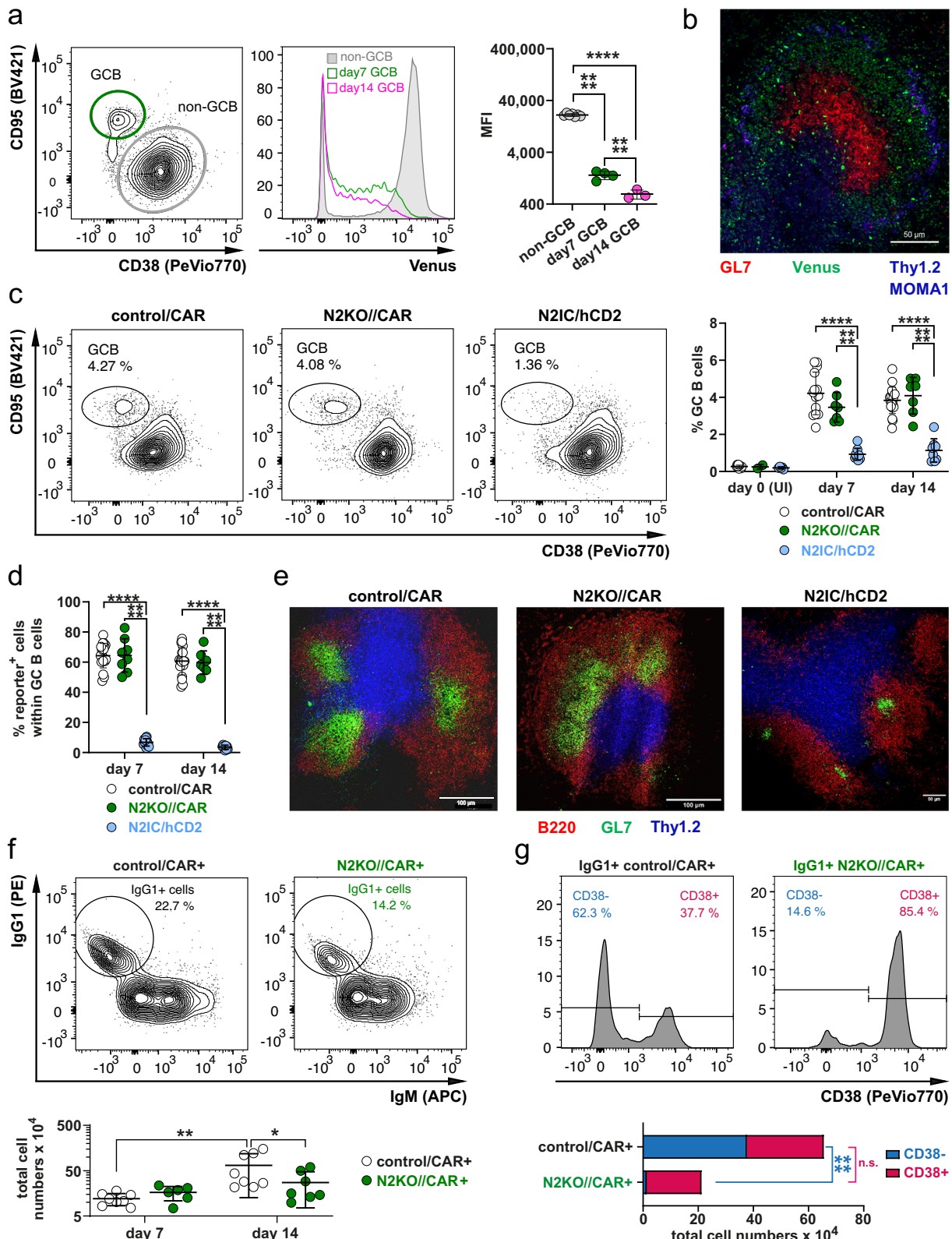

of this population displayed a MZB cell phenotype based on their CD21/CD23 surface expression (Fig. 6c, d). The frequencies of these cells increased over time post-immunization (Fig. 6d, Supplementary Fig. 9). Concurrently, these cells increased their size (FSC) and upregulated IgM, CD1d and CD38 (Fig. 6e), consistent with the characteristic MZB cell phenotype. Moreover, the N2IC-expressing cells exhibited migration towards the marginal zone (Fig. 6f). Taken

together, our findings suggest that Notch2 signaling in antigen-activated FoB cells not only promotes their entry into PC differentiation, but also drives their differentiation into MZB cells.

To investigate whether MZB cell differentiation also occurs physiologically during TD-immune responses in B cells with normal Notch responsiveness, we examined the spleen of control/CAR mice after TD-immunization (Fig. 7a). Intriguingly, we observed the presence of Cre-

**Fig. 3 | Notch signaling is downregulated in most Germinal Center B cells but is re-induced in some Light Zone-GCB cells. a** FACS plots are pre-gated on B220$^+$ B lymphocytes and show the gating of germinal center B (GCB) (green gate) and non-GCB cells (gray gate). The histogram shows a representative overlay of the H2B-Venus expression in non-GCB cells (gray) and in GCB cells 7d (green) and 14d (pink) post-immunization (p.i.). The graph summarizes Median Fluorescence Intensity (MFI) of H2B-Venus in the analyzed populations. D7: $n = 4$ mice (1 male (m), 3 females (f)), d14: $n = 3$ mice (1 m, 2 f). After log transformation of data, an ordinary one-way ANOVA with Tukey´s multiple comparison test was performed (****$p < 0.0001$). **b** Splenic sections were stained for GCs (GL7$^+$; red), T cells (Thy1.2$^+$; blue) and metallophilic macrophages (Moma1$^+$; blue) d7 p.i. Venus signals appear in green. Scale bar represents 50 µm. $n = 2$ mice. **c** Representative FACS plots for GCB cells (CD95$^+$CD38$^{low}$) d7 p.i. **c, d** The graphs compile percentages of total (**c**) and reporter$^+$ (**d**) GCB cells in the spleen from control/CAR (white), N2KO//CAR (green) and N2IC/hCD2 (blue). Mouse numbers: d0 unimmunized mice (UI) $n = 10$ (6 m, 4 f), $n = 2$ (2 m), $n = 5$ (3 m, 2 f); d7: $n = 14$ mice (7 m, 7 f), $n = 7$ (2 m, 5 f), $n = 9$ (6 m, 3 f); d14: $n = 20$ (11 m, 9 f), $n = 7$ (6 m, 1 f), $n = 8$ (5 m, 3 f) for control/CAR, N2KO//CAR, N2IC/hCD2 mice, respectively. An ordinary two-way ANOVA with Tukey´s multiple comparison test was performed (****$p < 0.0001$) in (**c**) and an ordinary two-way ANOVA with Sidak´s multiple comparison test in (**d**) (****$p < 0.0001$). **e** Splenic sections from mice 7d p.i were stained for GCB cells (GL7$^+$; green), B cells (B220$^+$; red), and T cells (Thy1.2$^+$; blue). Scale bars represent indicated lengths. $n = 4$ mice per genotype. **f** Exemplary gating of IgG1$^+$ cells in the spleen d14 p.i.: The graph below depicts the total numbers of splenic IgG1$^+$ lymphocytes in control/CAR (white dots) and N2KO//CAR mice (green dots). After log transformation of data, an ordinary two-way ANOVA with Sidak´s multiple comparison test was performed (*$p = 0.0408$; **$p = 0.0028$). **c, d, f** The horizontal bar and the error bars show the mean value and the SD. **g** The histograms show the exemplary gating for CD38$^+$ and CD38$^-$ cells within the B220$^+$IgG1$^+$IgM$^-$ gate of reporter$^+$ cells (**f**). The bar chart below summarizes the total cell numbers of CD38$^+$ (red bar) and CD38$^-$ IgG1$^+$ (blue bar) cells. After log transformation, an ordinary two-way ANOVA with Sidak´s multiple comparison test was performed (****$p < 0.0001$). Mice numbers for (**f, g**): N2KO//CAR d7: $n = 6$ (2 m, 4 f), d14: $n = 7$ mice (6 m, 1 f); control/CAR mice d7: $n = 8$ mice (5 m, 3 f), d14: $n = 8$ mice (4 m, 4 f). Source data from (**a**), (**c**), (**d**), (**f**), and (**g**) are provided as a Source Data file.

---

reporter$^+$CD23$^{low}$CD21$^{high}$ cells, which gradually increased in frequency over time following immunization, ranging between 2–10% throughout the observation period (Fig. 7a and b).

Notably, these Cre-reporter$^+$CD23$^{low}$CD21$^{high}$ B cells were virtually absent in immunized N2KO//CAR mice, indicating that their generation is dependent on Notch2 (Fig. 7a, b). The newly generated Cre-reporter$^+$ MZB cells displayed a characteristic MZB cell phenotype (Supplementary Fig. 10a). The percentage of NP-specific cells was comparable within the Cre-reporter$^+$ MZB and memory compartments, while it was higher in FoB cells (Fig. 7c and Supplementary Fig. 10b). From day 14 to day 26, the percentages of NP$^+$ cells increased significantly in the MZB cell compartment, whereas they remained relatively constant in FoB and memory B cells (Fig. 7c). These data suggest that with progressive time post-immunization Cre-reporter$^+$NP$^+$ cells accumulate in the MZB cell compartment. To assess the contribution of CAR$^+$ MZB cells in recall immune responses, we performed booster immunizations in N2KO//CAR mice, lacking CAR$^+$ MZB cells, 35 days after the first immunization and analyzed their immune response 7 days later in comparison to control mice. NP-specific IgM and IgG1 serum titers were lower in N2KO//CAR compared to control/CAR mice (Fig. 7d), suggesting that the CAR$^+$ MZB cells contribute to recall immune responses by de novo PC production. However, further analyses are still required as defects in the memory responses of Notch2-deficient B cells could also contribute to the decreased serum titers.

## Quantitatively mapping of B cell differentiation during an immune response reveals bifurcation of FoB cells into GCB and MZB cell fates

To gain insights about the precursor cells of CAR$^+$ MZB cells, we produced correlation plots of CAR expression kinetics in the FoB, MZB, and GCB cell subsets in the course of the TD-immune reaction. We employed a sequential gating strategy to determine the frequencies of CAR-expressing B cells among the total GCB cells (CAR$^+$B220$^+$CD38$^{low}$CD95$^{high}$), FoB cells (CAR$^+$B220$^+$CD38$^+$CD95$^{mid/low}$CD23$^+$CD21$^{low}$) and MZB cells (CAR$^+$B220$^+$CD38$^+$CD95$^{mid/low}$CD23$^{low}$CD21$^{high}$). An example of this hierarchical gating is depicted in Supplementary Fig. 11a, b. The correlation analysis revealed that the rise and decline in CAR$^+$ GCB cell numbers mapped linearly with the accumulation of CAR-expressing FoB cells (Fig. 8a), underlining that GCB cells are mainly derived by the activation of FoB cells. We did not find any indication of a precursor-progeny relationship between CAR$^+$ MZB and either CAR-expressing FoB or GCB cells, as their CAR expression kinetics correlated very weakly (Fig. 8b). Therefore, to uncover the developmental origins of CAR$^+$ MZB cells, we next integrated the data from the immunization experiments into a deterministic mathematical modeling strategy. We hypothesized that CAR$^+$ MZB cells emerge either from FoB cells (branched pathway, Fig. 8c) or from GCB cells (linear pathway, Fig. 8d), in addition to potential activation of CAR$^-$ MZB cells by the TD-antigen, which results in CAR expression on these MZB cells. We also defined a 'null model' in which all CAR$^+$ MZB cells are generated only by the activation of CAR$^-$ MZB cells (Fig. 8e), without any influx from FoB and GCB cells.

The simplest forms of branched, linear, and null models assume neutral dynamics for the birth-loss of GCB and MZB cells i.e., the influx of new cells into the compartment and their net loss (defined as the balance between division and true loss by death and differentiation) remain unchanged over time. To model the effect of antigen availability on B cell dynamics, we considered time-dependent variation in the rate of influx of FoB cells into the GCB cell compartment ($\alpha(t)$) and in the rate of upregulation of CAR expression on MZB cells ($\beta(t)$). In this 'time-varying influx' model, the differentiation of FoB cells into CAR$^+$ MZB cells ($\mu$) remains unchanged with time. Additionally, we considered another extension of the neutral model in which the net rate of loss of GCB cells varies with time ($\delta(t)$), while the loss rate of CAR$^+$ MZB cells ($\lambda$) remains constant. Model specifics are described in further detail in the Materials and Methods.

We fitted each model simultaneously to the time-courses of counts of CAR$^+$ MZB cells and CAR$^+$ GCB cells from control/CAR mice (red dots in Fig. 9a) and N2KO//CAR mice (blue dots in Fig. 9a), using the Bayesian statistical approach and assessed the relative support for each model using the leave-one-out (LOO) cross validation, calculated as the model weight 'W' (estimates in Table 1a). Model weight calculation is described in detail in Supplementary Note 1. We consider that neither FoB nor GCB cells contribute to the generation of CAR$^+$ MZB in N2KO//CAR mice (i.e., $\mu = 0$), since activation induces Notch2 deletion in B cells in these mice. We also assumed different loss rate of CAR$^+$ MZB cells ($\lambda'$) in N2KO//CAR mice. The influxes into the CAR$^+$ compartments of GCB and MZB cells were defined using the empirical descriptions of the pool sizes of total FoB cells and CAR$^-$ MZB cells (Supplementary Fig. 12 and details in Supplementary Note 2).

We found that the combination of the branched pathway with time-varying influx model received the strongest statistical support from the data ($W = 79\%$, Table 1a). We also explored a version of time-varying influx model in which the rate of differentiation of FoB into CAR$^+$ MZB varied with time ($\mu(t)$), which received poorer statistical support ($W = 17\%$), in One-on-one comparison to the best-fitting model ($W = 83\%$).

Our best-fitting model nicely captured the rise and fall in CAR$^+$ GCB cell numbers and described the persistence of CAR$^+$ MZB cells for over a month post-immunization in control/CAR mice (Fig. 9a). The model estimates that activation of CAR$^-$ MZB cells contributes

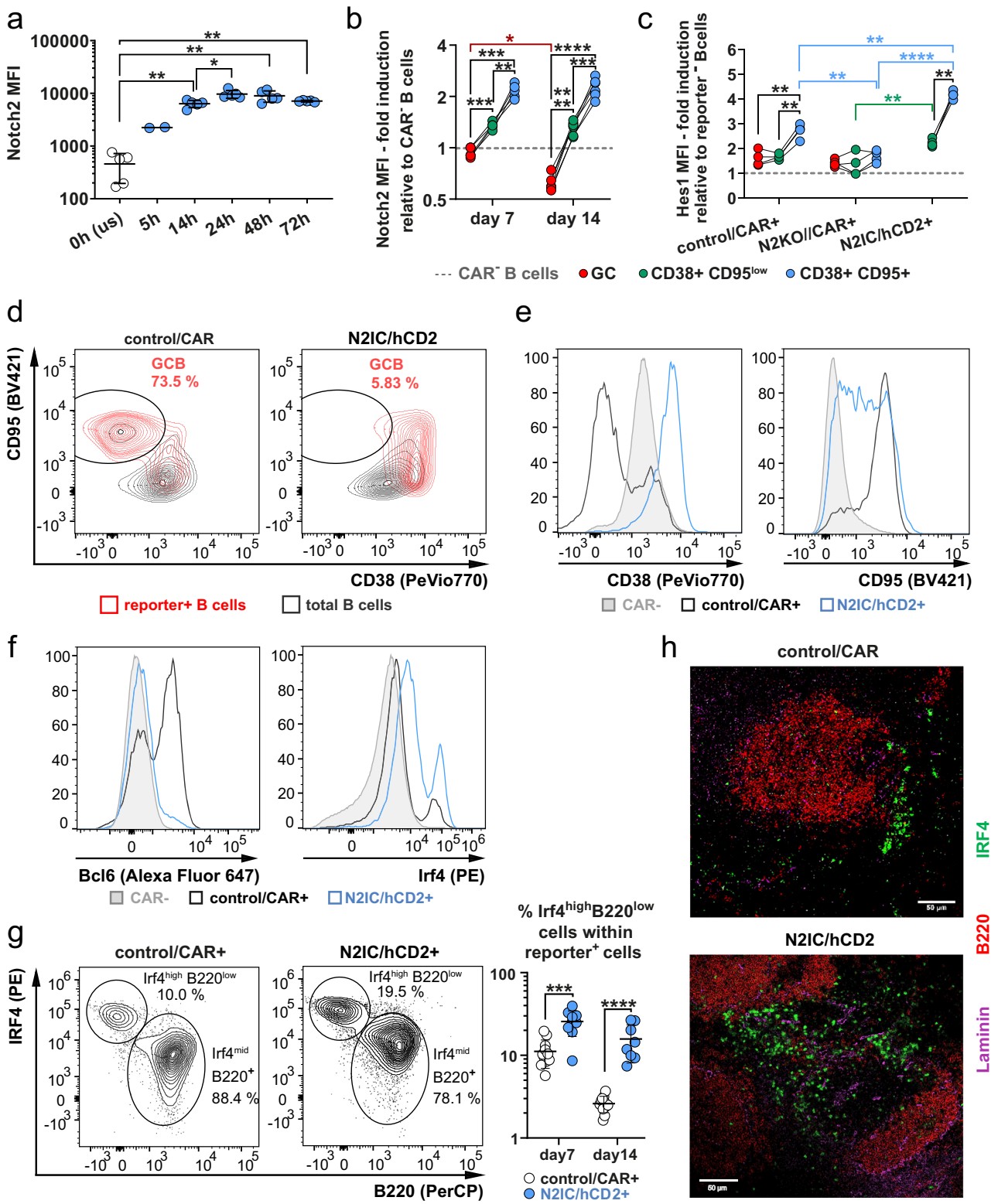

approximately 25% to the CAR[+] fraction in the MZB cells at day 4, which further declines steadily to near-zero by week 3 post-immunization (Fig. 9b and Table 1b). The numbers of CAR[+] MZB cells are primarily sustained by the differentiation of antigen-activated FoB cells (Fig. 9b and Table 1b), highlighting the importance of Notch2 signaling in their development and maintenance. Our model predicts that CAR[+] MZB cells are lost twice as rapidly in Notch2-deficient mice as compared to the control/CAR mice with functional Notch2 signaling (Fig. 9c and Table 1b). The shorter half-lives of CAR[+] MZB clones in addition to the

absence of influx from CAR[+] FoB cells, explains their substantially diminished pool sizes in N2KO//CAR mice (Fig.9a, blue lines and dots). Overall, our modeling and experimental results reveal an instructive role of Notch2-mediated signals in determining B cell diversification patterns during immune responses.

## Discussion

The conventional view of B cell development suggests that in mice, MZB cells emerge from transitional T2 cells, which bifurcate into

**Fig. 4 | Notch2 surface expression and signaling is enhanced in activated pre-Germinal Center B cells. a** Median Fluorescence Intensity (MFI) of Notch2 surface expression on purified splenic Follicular B (FoB) cells after α-IgM+α-CD40 stimulation. $n = 5$ mice (2 males (m), 3 females (f)), except for 5 h $n = 2$ mice (2 m) (unstimulated (us)): white dots; stimulated cells after different time points: blue dots. The horizontal bar and the error bars show the mean value and the standard deviation (SD). Data were log-transformed. Brown-Forsythe and Welch ANOVA with Dunnett´s T3 multiple comparisons test (*$p = 0.037$ (0h–5h); **$p = 0.0014$ (5h–14h); *$p = 0.037$ (14h-24h); **$p = 0.021$ (0h–48h); **$p = 0.058$ (0h–72h)). **b, c** Gating of Germinal Center B (GCB) (CD38−CD95+), non-GCB (CD38+CD95low), and activated FoB/memory B cells (CD38+CD95high) is depicted in Supplementary Fig. 6b. **b** Fold induction of Notch2 cell surface expression Median Fluorescence intensity (MFI) in GCB cells (red), CD38+CD95low cells (green), CD38+CD95+ cells (blue) from control/CAR mice, relative to its expression levels in CAR− B cells (MFI set to 1, depicted by the dotted line). d7 $n = 5$ (2 m, 3 f), d14 $n = 6$ mice (2 m, 4 f). For subpopulation comparisons within each time point, a repeated measures (RM) two-way ANOVA with Sidak´s multiple comparisons test was applied (*$p = 0.0119$ d7 ***$p = 0.0002$, ***$p = 0.0008$, **$p = 0.0046$; d14 ****$p \leq 0.0001$, ***$p = 0.0004$) and for comparisons between time points an ordinary two-way ANOVA with Sidak's multiple comparisons test (*$p = 0.0119$). **c** Fold induction of Hes1 (MFI) in the indicated populations of Cre-reporter+ cells (red dots:

GCB cells; green dots: CD38+CD95low; blue dots: CD38+CD95+) relative to its expression levels in Cre-reporter− B cells (MFI set to 1, depicted by the dotted line) $n = 4$ mice for each genotype: control/CAR: 2 m, 2 f; N2KO//CAR: 2 m, 2 f; N2IC/hCD2: 3 m, 1 f. After log transformation of data, a RM two-way ANOVA with Tukey´s multiple comparisons test was applied (left to right: **$p = 0.0077$, **$p = 0.001$, **$p = 0.0017$). **d** Overlays between Cre-reporter+ B cells (red) and total B220+ cells (black) are shown. Gates and percentages refer to the Cre-reporter+ GCB cells. **e, f** Histogram overlays depict the expression of the indicated markers in CAR− cells (filled gray); reporter+ cells from control/CAR mice (black line); reporter+ cells from N2IC/hCD2 mice (blue line). **d–f** Analyses are representative for $n = 14$ control/CAR and $n = 9$ N2IC/hCD2 mice. **g** Gating and percentages of Irf4highB220low cells within splenic Cre-reporter+ lymphocytes: d7: $n = 11$ (6 m, 5 f) and $n = 9$ (6 m, 3 f); d14: $n = 11$ (5 m, 6 f) and $n = 8$ (4 m, 4 f) control/CAR (white dots) mice and N2IC/hCD2 (blue dots) mice, respectively. The horizontal bar and the error bars show the mean value and the SD. After log transformation of data, an ordinary two-way ANOVA with Tukey´s multiple comparisons test was applied (***$p = 0.0002$, ****$p < 0.0001$). **d–g** All FACS plots are representative for d7 post-immunization. **h** Immunofluorescence analysis of splenic sections, stained for (pre-) plasmablasts and plasma cells (Irf4high; green), B cells (B220+; red) and marginal zone (MZ) sinus (Laminin; purple). Scale bar represents 50 μm, $n = 3$ mice per genotype. Source data from (**a–c**), (**e**), and (**g**) are provided as a Source Data file.

mature FoB cells and MZB cells in the spleen, dependent on Notch2 signaling[6]. However, recent studies demonstrated that FoB cells can also differentiate into MZB cells[11,14], but it remains unresolved under what circumstances this occurs during normal physiological processes. In this study, we demonstrate that the transition from FoB to MZB cells can be triggered by B cell activation. Our experimental data and mathematical modeling suggest that integration of Notch2, BCR, and CD40 signaling triggers a binary switch in antigen-activated B cells leading to the development of GCB or MZB and plasma cells. Our findings suggest, that activated B cells that downregulate Notch signaling and Irf4 expression develop into GCB cells, while sustained and enhanced Notch signaling leads to the development of MZB cells or PC cells, likely depending on their Irf4 levels[29–31]. Our analyses based on Notch-reporter transgenes[24] and endogenous Notch-target genes demonstrated that except GCB cells and PCs, all immature and mature B cell populations receive a basal Notch signal. Basal ongoing Notch2 signaling in FoB cells is further corroborated by lower expression of classical Notch-target genes in FoB cells in Notch2-deficient mice (Fig. 1c) and in FoB cells from mice treated with a Notch2-inhibitor[32] in comparison to controls. A relatively high stability of H2B-Venus in immature and FoB cells from CBF:H2B-Venus mice may result in its gradual increase during B cell development, if B cells receive recurring Notch signals or if Notch-IC remains in the nucleus for an extended time. Pre-MZB and MZB cells show the highest H2B-Venus expression, supporting the hypothesis that a strong Notch2 signal promotes MZB cell development. Surprisingly, ~50% of MZB cells exhibit downregulation or complete loss of the H2B-Venus signal. We assume this is linked to their intrinsic ability to proliferate and to differentiate into PCs. In support of this, the H2B-Venuslow MZB subset exhibited higher percentages of Ki-67+ cells and elevated levels of Irf4 and Blimp1, indicating increased proliferation and a greater tendency to differentiate into PCs in comparison to H2B-Venushigh MZB and FoB cells. Our hypothesis gains further support from our observation that Lipopolysaccharide (LPS)-treatment of B cells in vitro leads to a substantial loss of H2B-Venus, attributed to both proliferation and differentiation. The decline of H2B-Venus during proliferation is likely due to dilution, while its decrease during differentiation may result from extensive genome re-organization processes in plasmablasts[33], potentially releasing H2B-Venus from the chromatin. Additionally, the H2B-Venus reduction may be associated with a faster degradation of H2B-Venus in plasmablasts and plasma cells compared to other mature B cell populations. Earlier data already demonstrated a variable H2B-

Venus protein stability in different tissues, with a very short half-life in certain cases[24].

It was striking that both H2B-Venushigh and H2B-Venuslow MZB cells had higher Hes1 levels than FoB cells. In contrast to Notch-target genes, the Notch-reporter functions as a direct indicator of Notch signaling, while Notch-target genes are regulated by various pathways. This distinction may explain the faster downregulation of H2B-Venus in comparison to Hes1 in MZB cells, which did not receive a fresh Notch signal within a certain time.

The fate of the H2B-Venuslow cells remains uncertain. They may undergo cell death and be replaced by newly immigrating MZB cells, or they may receive a new Notch2 signal and cycle back to the H2B-Venushigh state. Notch2 signaling is essential for the maintenance of MZB cells[31,32], but the source of this maintenance signal has yet to be determined. Dll-1 is only expressed within the follicle by Follicular Dendritic cells (FDC) or by Marginal Reticular cells that are located on the marginal zone sinus on the inner side[20], suggesting that MZB cells have to migrate into the follicle or at least at the marginal zone sinus border to refresh their Notch2 signal[34,35]. Alternatively, the maintenance signal may be conveyed by another ligand, such as Dll-4, which is strongly expressed by endothelial cells from the marginal zone sinus[19].

Furthermore, our data re-confirmed the findings from previous studies demonstrating that strong and sustained Notch signaling prevents the formation of germinal centers[15,25]. Bcl6, the main regulator of the GC reaction, actively represses the transcription of Notch2 and other Notch pathway genes[25,36]. In this study, we show that vice versa sustained Notch2 signaling inhibits the upregulation of Bcl6 and instead results in upregulation of Irf4. Inhibition of Bcl6 expression may be mediated by the Notch-target gene Hes1, which has been shown recently to repress Bcl6 expression and negatively regulate GC formation[37]. We found that Irf4high cells initiate PC differentiation, while Irf4med cells differentiate into MZB cells. Although Notch2 signaling is downregulated in GCB cells[25], our H2B-Venus data suggest that some LZ-B cells may receive a Notch2 signal, likely provided by T follicular helper (Tfh) cells and/or FDCs, which are known to express high levels of Notch ligands Dll-1 and Jagged-1[38,39]. Notch2 signaling appears to positively regulate the numbers of IgG1-switched cells in the GC. In vitro studies from the Allman group already suggested a role of Notch signaling in the generation of IgG1-switched cells[40]. Switching occurs mainly in pre-GCB cells[41]. Both IgM+ and IgG1+ B cell blasts differentiate into GCB cells, where they proliferate but do not undergo further class-

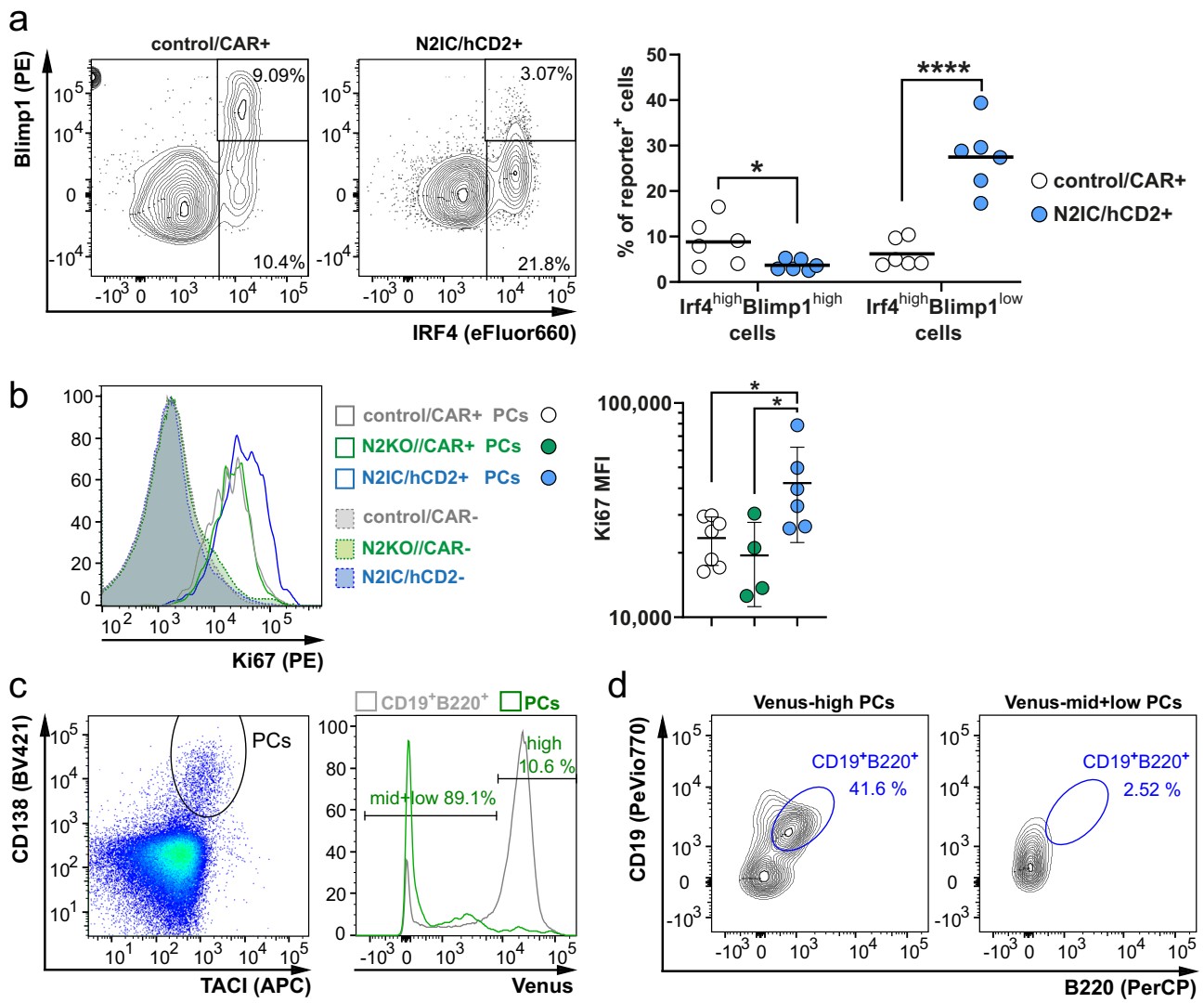

**Fig. 5 | Notch2 signaling initiates plasma cell differentiation but must be downregulated to enable terminal differentiation. a** Splenocytes were pre-gated on living Cre-reporter⁺ lymphocytes. The Irf4^high fraction was divided into a Blimp1^low pre-plasmablast and early plasmablast (pre-PB and early PB) and a Blimp1^high fraction (late PB and plasma cells (PCs)). Percentages of Blimp1^high and Blimp1^low cells in the Cre-reporter⁺Irf4^high fraction are summarized in the graph, $n = 6$ control/CAR mice (3 males (m), 3 females (f), (white dots), and $n = 6$ N2IC/hCD2 mice (3 m, 3 f) (blue dots). The horizontal bars show the mean values. After log transformation of data, unpaired two-tailed *t*-tests for each subpopulation were performed (*$p = 0.0248$, ****$p < 0.0001$). **b** Splenocytes were gated on living Cre-reporter⁺Irf4^highB220⁻ cells (PC) and analyzed for Ki-67 expression. The histogram overlay shows the Ki-67 levels in the indicated populations (filled histograms with dotted lines: reporter⁻cells; unfilled histograms with solid lines: reporter⁺ PCs) and genotypes (gray: control/CAR; green: N2KO//CAR; blue: N2IC/hCD2) at d7 post-

immunization (p.i.). The graph summarizes the Median Fluorescence Intensity (MFI) of the Ki-67 expression in PCs from control/CAR: white dots, $n = 7$ mice (2 m, 5 f), N2KO//CAR: (green dots), $n = 4$ mice (2 m, 2 f) and N2IC/hCD2: blue dots, $n = 6$ mice (3 m, 3 f). The horizontal bar and the error bars show the mean value and the standard deviation. Data were log-transformed and applied to an ordinary one-way ANOVA with Tukey´s multiple comparisons test (left to right: *$p = 0.0435$, *$p = 0.0145$). **c** Splenocytes from CBF:H2B-Venus mice were pre-gated with a large lymphocyte gate and analyzed for PCs (TACI⁺CD138⁺). The histogram shows an overlay of H2B-Venus in PCs (green) and B220⁺CD19⁺ B cells (gray). Gates and percentages refer to the green histogram of the PCs. **d** Contour plots depict the CD19/B220 expression in H2B-Venus^high and H2B-Venus^mid+low PCs (TACI⁺CD138⁺). **c**, **d** The Flow cytometry analyses were performed at d7 p.i. and are representative for $n = 4$ mice. Source data from (**a**) and (**b**) are provided as a Source Data file.

switch recombination. Around 10 days post-immunization IgG1-switched cells start to be positively selected at the expense of IgM⁺ cells[42]. Our combined data showing that the number of IgG1⁺ cells is similar between N2KO and control mice at day 7, but significantly diminished in N2KO mice at day 14 of the GC reaction, together with the stronger H2B-Venus expression in a portion of LZ-B cells, suggest that Notch2 signaling in LZ-GCB cells may regulate the positive selection and expansion of IgG1⁺ B cells.

We also provide evidence that Notch signaling plays a dual role in PC differentiation by initiating the process, while hindering the terminal differentiation. Thereby, Notch signaling may keep plasmablasts in

a proliferative stage by inhibiting the further upregulation of Blimp1[43-47]. Accordingly, the reduced PCs in N2KO mice at day 7 after immunization may be due to an impaired proliferation of plasmablastic cells in the early immune response. Previously, a role of Notch signaling in PC differentiation was mainly attributed to an interaction between Dll-1 and Notch1[38,48,49]. Therefore, the relatively slight effect of Notch2 ablation on PCs could be due to a redundancy of Notch1- and Notch2 signaling in PC differentiation, with the main contribution being at Notch1.

Mathematical modeling of data derived from our immunization studies in antigen-inducible Cre-reporter mice identified a Notch2-

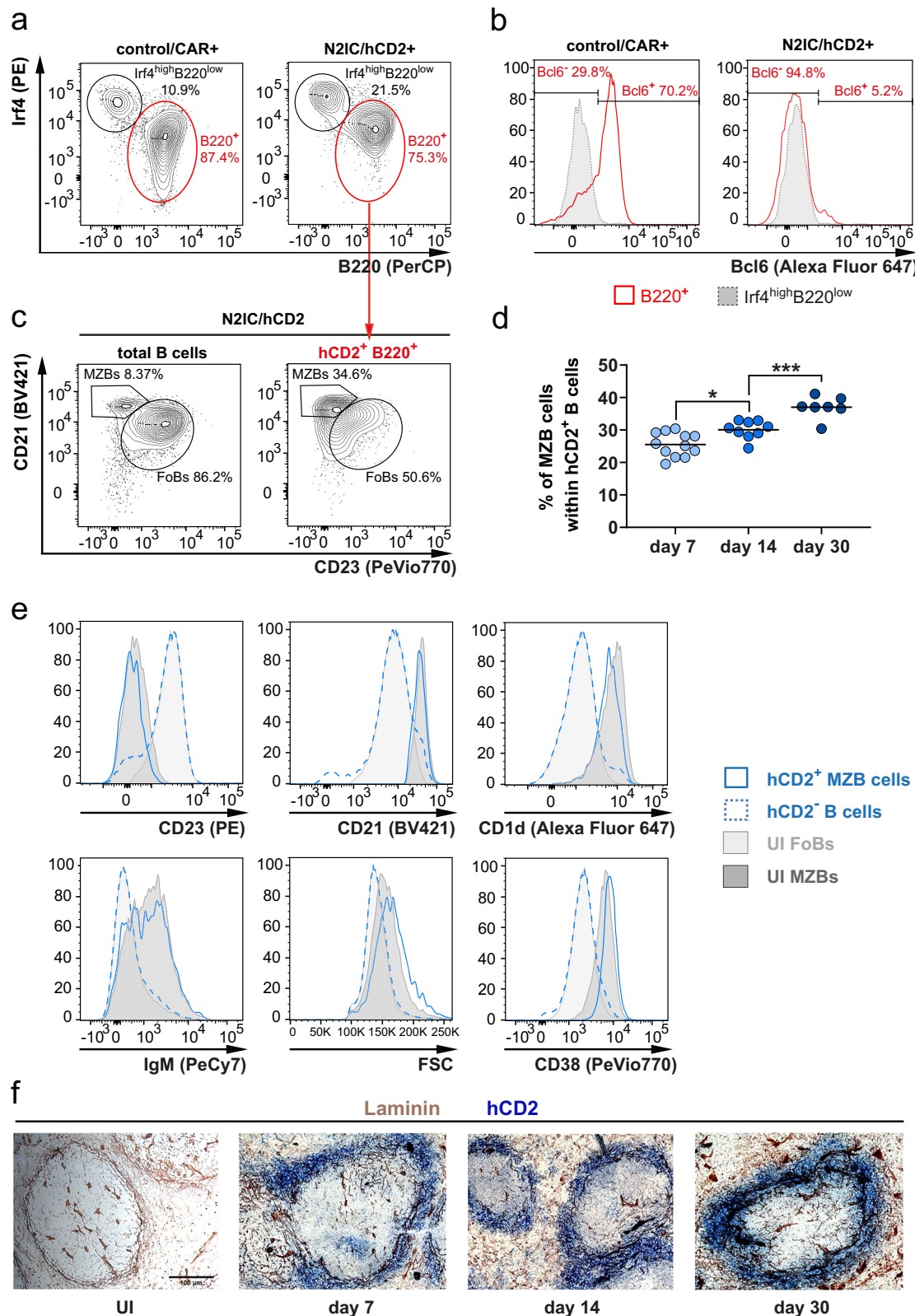

dependent differentiation route, in which MZB cell generation during an immune response is primarily sustained by differentiation of activated non-GC FoB cells. Earlier studies already hinted at an antigen-dependent generation of MZB cells[50–53]. However, it is still elusive whether these MZB cells originate from GC or pre-GC B cells. Studies showing that FoB and MZB cells are clonally related in rats[54] and that memory-like antigen-specific CD80+/CD21high cells in the MZ are less mutated than CD80+/

CD21low cells[55] support our modeling data, which estimated that the antigen-specific CAR+CD21high MZB cells are mainly generated from antigen-activated pre-GCB cells. It is possible that these MZB cells generated during immune responses adopt a memory-like phenotype. In support of this hypothesis are previous studies showing that antigen-induced memory B cells are equally distributed between the MZ and the B cell follicles, even at later time points after immunization[55] and that

**Fig. 6 | N2IC/hCD2⁺ B cells differentiate into Marginal Zone B cells with the correct surface phenotype and splenic localization. a** Cre-reporter⁺ lymphocytes were separated into B220^low Irf4^high (gate with the black line) and B220⁺ cells (gate with the red line). **b** Histograms show overlays of the Bcl6 expression in Cre-reporter⁺ B220⁺ (red line) and Cre-reporter⁺ B220^low Irf4^high cells (filled gray). **c** Gating of Marginal Zone B (MZB) cells (CD21^high CD23^low) and Follicular B (FoB) cells (CD23^high CD21^low) in N2IC/hCD2 mice. The left plot is gated on total B220⁺ cells, and the right plot on hCD2⁺ B220⁺ Irf4^mid cells. **a**–**c** FACS plots are representative for d7 post-immunization (p.i.). n = 7 control/CAR and n = 4 N2IC/hCD2 mice. **d** Graphical summary of the percentages of CD21^high CD23^low MZB cells within Cre-reporter⁺ cells from (**c**) at indicated time points. d7 n = 12 mice (6 males (m), 6 females (f)), d14 n = 9 mice (5 m, 4 f) and d30 n = 7 mice (6 m, 1 f). An ordinary one-way ANOVA with

Tukey´s multiple comparison test was performed (*p = 0.0119, ***p = 0.0009). **e** Histogram overlays of the expression of indicated markers within Cre-reporter⁺ MZB cells (solid blue line) and total splenic Cre-reporter⁻ B cells (dotted blue line) from N2IC/hCD2 mice at d7 post-immunization (p.i.), compared to total MZB (filled histogram dark gray) and FoB cells (filled histogram bright gray) from unimmunized (UI) N2IC/hCD2 animals. Analyses are representative for n = 9 immunized N2IC/hCD2 mice and n = 4 UI N2IC/hCD2 mice. **f** Chromogenic immunohistochemical analysis for the localization of hCD2⁺ cells at the indicated time points. Splenic sections were stained for hCD2 (blue) and for basement membranes of endothelial cells lining the marginal zone (MZ) sinus with Laminin (brown). Scale bar represents 100 μM and applies to all images. n = 3 mice per time point. Source data from (**d**) are provided as a Source Data file.

memory B cells with a MZB cell expression profile are detected in the spleen after viral infection[56]. On the other hand, we cannot exclude that some of the CAR⁺ MZB cells are directly generated from memory B cells. This would lead to a CAR⁺ MZB cell compartment consisting of both somatically mutated and non-mutated 'antigen-experienced' MZB cells, similar to what has been described in humans[57]. Further investigation is required to understand the dynamics of memory B cell dispersion between MZB and FoB cell compartments and to examine differentiation trajectories within antigen-experienced B cell lineages. For instance, tracking dynamics of MZB cells derived from purified donor FoB or GCB cells in congenic recipient mice may be able to untangle their individual contributions to MZB generation during immune responses. We will explore this direction in a future study, but in the scope of this work, we show results from a theoretical simulation of this transplantation experiment (Supplementary Fig. 13), generated using the parameters derived from fitting the branched and linear models to the data from control/CAR mice. The details of the simulation strategy are outlined in the Supplementary Note 3.

In addition, it will be interesting to investigate the mutation rate and clonal composition of the CAR⁺ MZB cell compartment. In humans, the MZB compartment appears to change during aging[58]. So, it will be interesting to explore whether the clonal composition and frequencies of hypermutated MZB cells change during aging in mice and whether the ratio of naïve and antigen-experienced MZB cells can be shifted in favor of antigen-experienced MZB cells through frequent antigenic exposures.

## Methods
We confirm that the research compiles with all relevant ethical regulations. All animal experiments were performed in compliance with the German Animal Welfare Law and were approved by the Institutional Committee on Animal Experimentation and the Government of Upper Bavaria.

### Mouse models
CBF:H2B-Venus reporter mice[24] were purchased from The Jackson Laboratory (JAX stock #020942), backcrossed into the Balb/c background and further maintained in house. Previously described R26/CAG-CARΔ1^StopF[27] and N2IC^flSTOP mice[15] were crossbred to the Cγ1-Cre strain[26] to generate R26/CAG-CARΔ1^StopF//γ1-Cre (control/CAR) and N2IC^flSTOP//γ1-Cre (N2IC/hCD2) mice, respectively. R26/CAG-CARΔ1^StopF mice carry the human coxsackie/adenovirus receptor CAR in the Rosa26 locus, preceded by a loxP-flanked STOP cassette. Cre recombinase activity results in an expression of a truncated version of the CAR Cre-reporter on the cell surface. In the N2IC^flSTOP mice, the conditional Notch2IC allele is directly coupled to the coding sequence for the human CD2 (hCD2) receptor through an internal ribosomal entry (IRES) site, preceded by a loxP-site-flanked STOP cassette in the Rosa26 locus. Cre recombinase-mediated excision ensures expression of both Notch2IC inside the cell and hCD2 on the cell surface. Lastly, Notch2^fl/fl mice[18] were mated to the R26/CAG-

CARΔ1^StopF//γ1-Cre animals to obtain the Notch2^fl/fl//R26/CAG-CARΔ1^StopF//Cγ1-Cre (N2KO//CAR) mice. Upon Cre-mediated recombination, exons 28 and 29 of the Notch2 gene, which code for the C-terminal part of RAM23 and the nuclear localization signal (NLS), and the STOP cassette upstream of the Car gene in the Rosa26 locus are excised. Thereby, Cre-recombined B cells express a truncated non-functional Notch2 receptor (because the following exons 30–33 of Notch2 are out of frame and are thus not translated) and the Cre-reporter CAR on the cell surface. All mice were on a pure Balb/c background, except for CBF:H2B-Venus reporter mice, which were backcrossed after purchase for 5 generations on a Balb/c background. The animals were kept in the laboratory animal rooms under specified pathogen-free (SPF) conditions with a 12/12-h light cycle. The animal rooms are fully air-conditioned, the set-points are set to 20–24 °C temperature and 45–65% humidity according to Annex A of the European Convention 2007/526 EC. The rooms are equipped with individually ventilated cage systems (e.g., Tecniplast Greenline GM 500, 501 cm² floor area, or BioZone-IVC type II, 370 cm² floor area). The maximum stocking densities comply with Annex III of Directive 2010/63/EU. The cages are equipped with laboratory animal bedding (wood fiber/wood chips, e.g., Lignocel Select Fine, SAFE). The animals were given sterile filtered water and a standard rodent diet (e.g., Altromin 1314) ad libitum. Control and experimental mice were bred separately but were kept in the same area of the animal facility. The mice were euthanized either by $CO_2$ or cervical dislocation. The QuietekTM apparatus was used for euthanasia with $CO_2$ to ensure that the $CO_2$ was supplied with the correct flow rate up to the correct filling volume.

Both male and female mice were used for the experiments, data were analyzed without consideration of the animals' sex. To ensure a more balanced distribution of both male and female animals throughout the study, control and analytical mice were sex-matched in experiments, whenever possible. A detailed overview of the sex and age of mice in each experiment is provided in the Source Data file and in the Figure legends.

### Mouse immunizations
To trigger TD-immune responses, 10- to 18- week-old mice were injected intraperitoneally with 100 μg of alum-precipitated 4-hydroxy-3-nitrophenylacetyl (NP)-chicken-gamma-globulin (CCG) (Biosearch Technologies, Novato, CA) in 200 μl sterile PBS (Gibco). Mice were analyzed at indicated time points after antigen injection. Respective unimmunized (UI) animals were taken down on the day of each experiment and were used as a day 0 time point. For the re-boost experiments examining secondary immune response, mice were injected with 50 μg NP-CCG solved in sterile PBS at day 35 after the first immunization and analyzed 7 days later.

### Antibodies used in this study
All details about the different antibodies used in Flow cytometry (FACS), Enzyme-linked Immunosorbent Assay (ELISA), Enzyme-linked

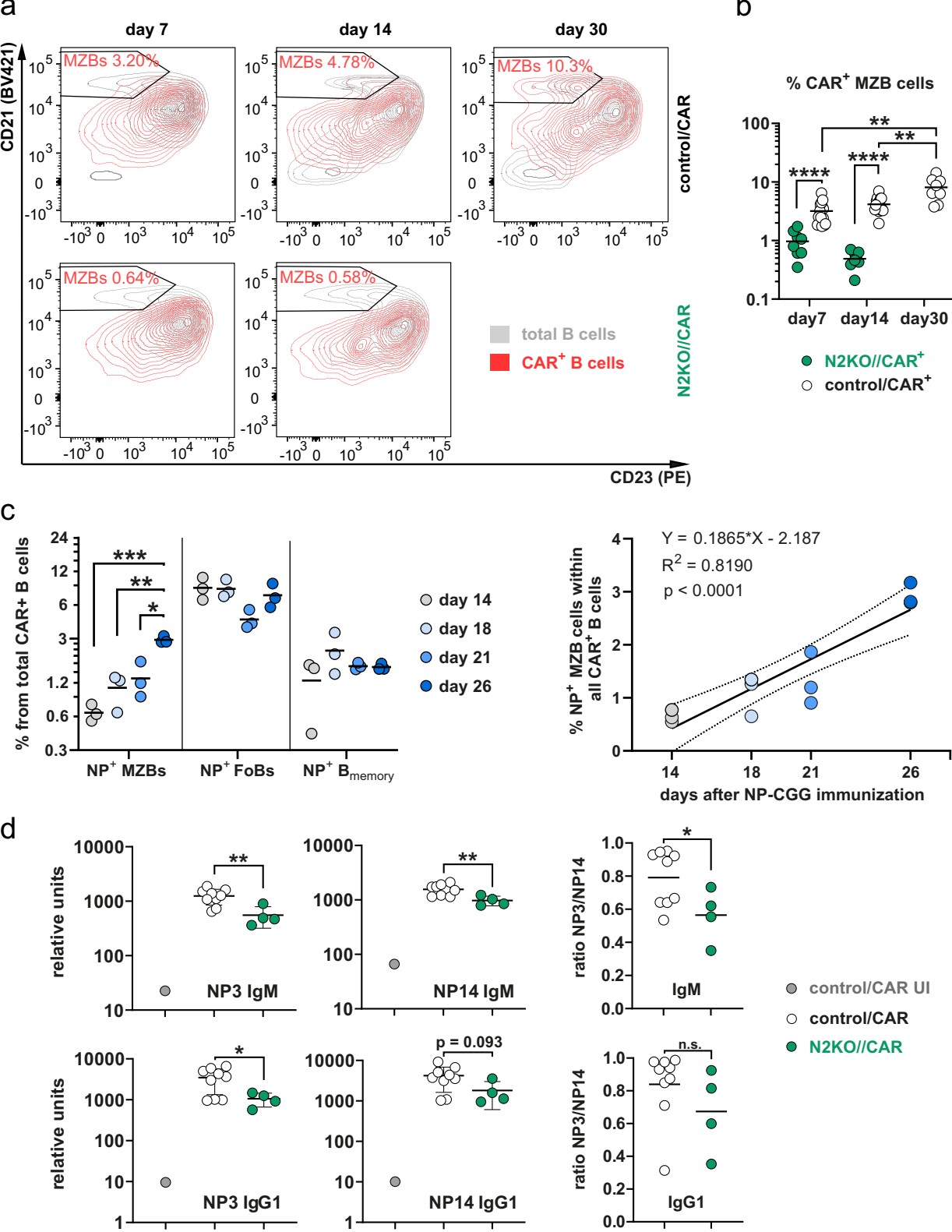

immunosorbent spot (ELISpot) assay, and histology are listed in Supplementary Table 1.

**Cell purification and in vitro cultures**

Naïve FoB cells of wildtype Balb/c mice were purified from splenic cell suspension using the MZB and FoB Cell Isolation Kit (Miltenyi Biotech) according to the manufacturer's instructions. Cells were cultured at a

density of $5 \times 10^5$ cells/well in flat-bottom 96-well plates in RPMI-1640 medium (Gibco), supplemented with 10% fetal calf serum (FCS, PAA Cell Culture Company), 1% L-glutamine, 1% non-essential amino acids, 1% sodium pyruvate and 50 μM β-mercaptoethanol. Except for FCS, all supplements were purchased from Gibco. An agonistic antibody against CD40 (2.5 μg/ml; eBioscience HM40-3) and an antibody specific for IgM (15 μg/ml; AffiniPure F(ab')2 goat-anti-mouse IgM, μ-chain,

**Fig. 7 | Notch2 signaling is necessary for the generation of Cre-reporter+ B cells with a Marginal Zone B cell phenotype. a** Representative flow cytometry analysis of Cre-reporter+ B cells (red) overlaid with total B220+ B cells (gray) in control/CAR and N2KO//CAR mice at the indicated time points. Cells are pre-gated on B220+ B lymphocytes. Numbers in the FACS plots indicate the percentages of Marginal Zone B cells (MZB) within Cre-reporter+ B cells. **b** The graph summarizes percentages of MZB cells within Cre-reporter+ cells from (**a**). Control/CAR mice (white dots): d7 $n = 16$ (8 males (m), 8 females (f)), d14 $n = 17$ (10 m, 7 f), d30 $n = 8$ (4 m, 4 f); N2KO// CAR mice (green dots): d7 $n = 8$ (3 m, 5 f), d14 $n = 7$ (6 m, 1 f). After log transformation, an ordinary two-way ANOVA with Tukey´s multiple comparison test was performed (****$p < 0.0001$, **$p = 0.0025$, **$p = 0.0026$). **c** Splenocytes were prepared from control/CAR mice. The graph illustrates percentages of 4-hydroxy-3-nitrophenylacetyl-specific (NP+) cells within the fraction of CAR+ cells in the indicated populations at the indicated time points p.i. (depicted as gray dots and different shades of blue as indicated). The gating strategy of the indicated

populations and the calculation of NP+ cells are described in Supplementary Fig. 10b. $n = 3$ mice for each time point d14: 3 m, d18: 3 f, d21: 3 f, d26: 2 m, 1 f. After log transformation, data were compared only within each cell population by an ordinary one-way ANOVA with Tukey´s multiple comparisons test (***$p = 0.0009$, **$p = 0.0093$, and *$p = 0.0291$). The right plot shows a simple linear regression analysis of the CAR+NP+ MZB cell data. Relevant parameters are listed in the graph. **d** NP-specific antibody titers of control/CAR (white dots) and N2KO//CAR (green dots) mice at d7 after secondary immunization with NP-CGG. The right graphs show the NP3/NP14 ratios of the respective isotypes. unimmunized control: $n = 1$ (1 m) mouse (gray dot); immunized control $n = 9$ mice (7 m, 2 f); immunized N2KO $n = 4$ mice (1 m, 3 f). ELISA data were log-transformed. Since only immunized genotypes were statistically compared, unpaired two-tailed t-tests were applied (IgM: **$p = 0.0033$, **$p = 0.0047$ and *$p = 0.0433$; IgG1: *$p = 0.0378$, $p = 0.093$ and $p = 0.246$). Source data from (**b**), (**c**), and (**d**) are provided as a Source Data file.

Dianova) were used as a combined stimulus for B cells, which were harvested at the indicated time points after addition of the stimulus and analyzed by flow cytometry after staining with an anti-Notch2-PE antibody (BioLegend). Dead cells were discriminated from living cells using a fixable live/dead cell staining kit (Invitrogen). For the plasmablast-differentiation assay, naïve B cells from CBF:H2B-Venus Notch-reporter mice were isolated from splenic cell suspension with the CD43 depletion B cell Isolation Kit (Miltenyi) according to the manufacturer's instructions. To monitor cell division, B cells were stained with CellTrace Far Red (Invitrogen) according to the manufacturer's protocol prior to the flow cytometry stainings. Cells were cultured at a density of $5 \times 10^5$ cells/well in flat-bottom 96-well plates as described above and were stimulated with 50 µg/ml LPS (Sigma). Cells were harvested at the indicated time points and analyzed by flow cytometry after staining with anti-CD138-BV421 and anti-B220-PerCP antibodies. Dead cells were discriminated from living cells using a fixable blue live/dead cell staining kit (Invitrogen).

## Histology

Murine splenic tissues were embedded in O.C.T. compound (VWR Chemicals, USA), snap frozen, and stored at −20 °C. Tissues were sliced with a cryostat with 7 µm thickness, mounted on glass slides, and stored at −80 °C for immunohistochemical (IHC) or immunofluorescence (IF) analysis.

For the chromogenic IHC, splenic sections were fixed with ice-cold acetone for 10 min, washed with PBS and blocked with 1% BSA, 5% goat serum in PBS. Samples were blocked using the Avidin/Biotin blocking kit (Vector), following the manufacturer protocol. Detection of hCD2-expression was done by staining with a biotinylated mouse-anti-hCD2 antibody and a rabbit-anti-Laminin antibody overnight at 4 °C in 1% PBS/BSA. Secondary antibody staining was done for 1 h at room temperature with streptavidin-coupled alkaline phosphatase and peroxidase-coupled anti-rabbit IgG in 1% PBS/BSA. Enzymatic chromogenic development was done with the AEC substrate and Blue AP substrate kits (Vector), following the manufacturer manual. Slides were embedded in Kaiser's Gelatin (Carl Roth) and analyzed using an Axioscope (Zeiss) microscope with an AxioCam MRc5 digital camera (Carl Zeiss GmbH)[14].

For the IF staining, sections were fixed for 10 min with PBS-diluted 3% Formaldehyde ph7 (Histofix, Carl Roth), rinsed in PBS, and rehydrated for 8 min in PBS + 50 mM $NH_4Cl$. For the visualization of the MZ, and B- cell zone, sections were blocked with 1% Bovine serum albumin (BSA), 5% rat serum, 5% chicken serum in PBS for 30 min, followed by Avidin/Biotin blocking (Vector) according to the manufacturer's protocol. Sections were stained with the primary antibodies (goat-anti-mouse IgM, anti-Moma1-Biotin) and the secondary antibody (chicken anti-goat IgG AF647) as well as Streptavidin AlexaFluor594. The antibodies and Streptavidin were incubated for 1 h at room temperature in 0.5% BSA/PBS.

To detect plasmablasts and PCs, sections were permeabilized and blocked for 20 min with 0.3% Triton X, 1% BSA, 5% goat serum in PBS. Primary (rat-anti-mouse Irf4, rabbit-anti-mouse Laminin) antibodies and secondary antibodies (goat-anti-rat AlexaFluor488, goat-anti-rat AlexaFluor647, goat-anti-rabbit Cy3) were incubated in 1% BSA/PBS for 1 h at room temperature. Fluorophore-coupled antibodies (anti-mouse B220-APC) were incubated for 2 h at room temperature, where appropriate. For the detection of germinal center (GC) structures, sections were blocked with 1% BSA, 5% rat serum in PBS for 30 min. Rat-anti-mouse GL7-FITC or Rat-anti-mouse GL7-APC antibody was incubated overnight at 4 °C. Directly coupled antibodies (Thy1.2-Biotin, Moma1-Biotin, and B220-APC) and Streptavidin AlexaFluor594 were incubated for 1 h at room temperature in 1% BSA/PBS. Slides were embedded in ProLong Glass Antifade (Invitrogen). Images were acquired on a TCS SP5 II confocal microscope (Leica), equipped with the Leica Application Suite X (LAS X) Software Version 4, and picture stacks were composed in ImageJ Version 1.50e[14].

## ELISA

NP-specific antibody titers in the sera were measured by ELISA, following the procedure described by[59,60]. NUNC plates (Nunc) were coated with NP3-BSA or NP14–BSA (10 mg/ml, Biosearch Technologies) in carbonate buffer (0.1 M $NaHCO_3$, pH = 9.5) and incubated overnight at 4 °C. Plates were washed with PBS, followed by blocking for 2 h with blocking buffer (1% milk powder in PBS for IgG1 ELISA, 5% milk powder in PBS for IgM ELISA). Serum samples were distributed in a 1:2 serial dilution in blocking buffer over 8 wells, starting with 100 µl/well of either 1:10 (IgM) or 1:100 (IgG1) diluted serum in the top row of the plate. The sera were incubated for 1 h at room temperature, followed by washing with PBS. For IgM ELISA, a directly coupled anti-IgM-horseradish peroxidase (HRP) antibody was diluted in blocking buffer and incubated for 1 h at room temperature. Afterwards the HRP signal was detected as described below. For IgG1 ELISA, a biotinylated rat-anti-mouse- IgG1 antibody was diluted in blocking buffer and incubated for 30 min at room temperature. Subsequently, plates were washed with PBS and incubated with streptavidin horseradish peroxidase Avidin D (Vector) diluted in blocking buffer. Following three washing steps with PBS and PBS-T, detection of the HRP signal was done with substrate buffer (1 tablet o-Phenylenediamine (Sigma, P-7288) + 35 mL substrate buffer (0.1 M citric acid, 0.1 M Tris (Sigma)) supplemented with 21 µl $H_2O_2$ (Sigma)). The absorbance was determined with a microplate ELISA reader (Photometer Sunrise RC, Tecan) at an optical density (OD) at 405 nm and data were acquired with the Infinite F200 PRO i-control software, Version 3.37. To correctly quantify the measured titers and compare different independent assays, internal standards consisting of pooled serums from six to eight NP-CGG immunized mice were used. A calibration curve was produced using the absorbance values measured in the serum standards, which was then used as a guideline

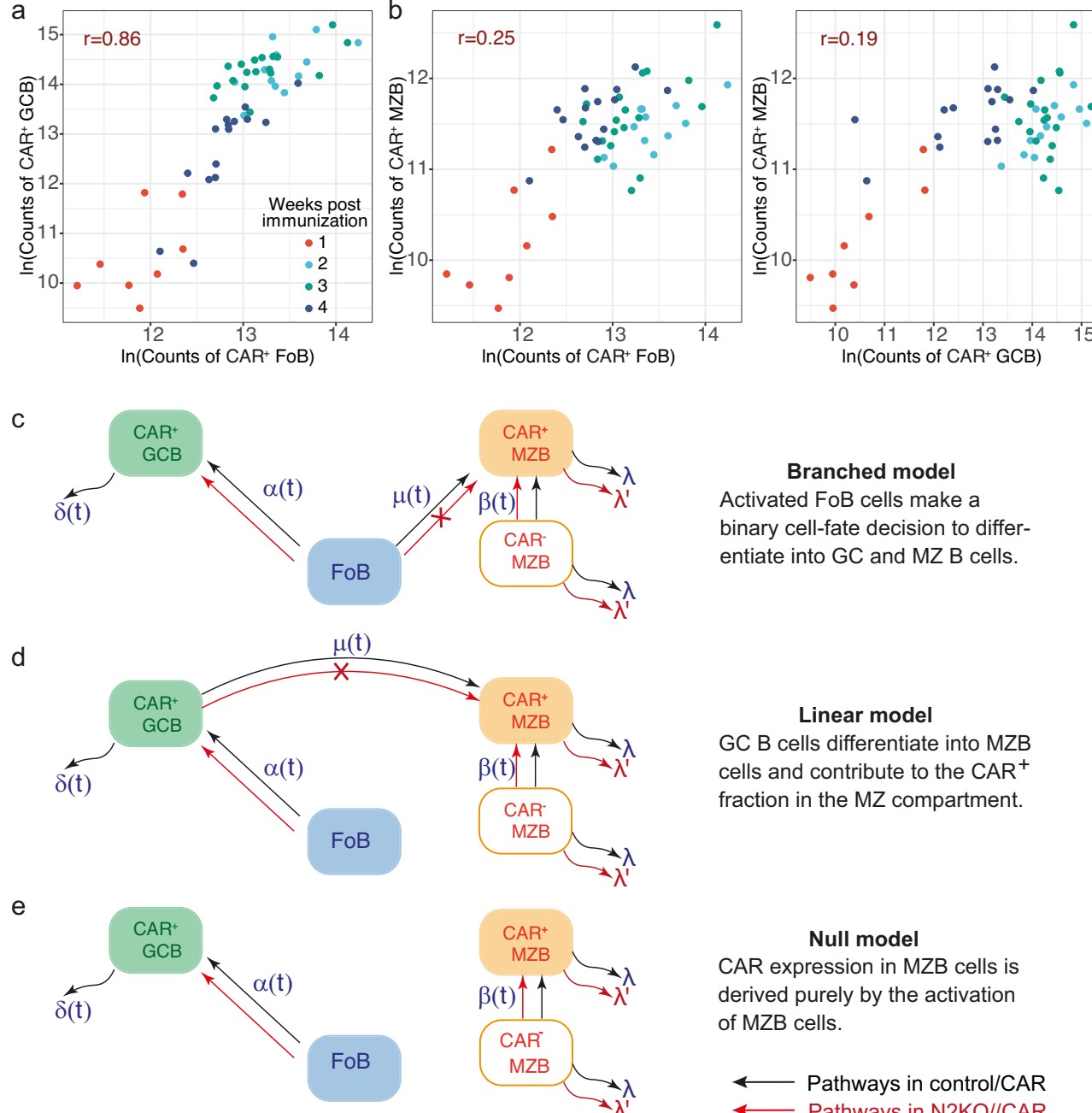

**Fig. 8 | Mathematical modeling of CAR expression dynamics in B cells during an ongoing TD-immune response. a, b** Correlation plots of CAR expression kinetics in Follicular (FoB), Marginal Zone B (MZB), and Germinal Center B (GCB) cell subsets. We compare the changes in the logarithmic (ln) counts of CAR⁺ B cells in each subset and show their correlation coefficient (*r*) on each plot. The sequential gating strategy to retrieve the frequencies of each cell population is depicted in Supplementary Fig. 11. The data are grouped in different bins based on the time of observation following TD-immunization, indicated by different colors in the plots: 1 week (red) (*n* = 8; 5 males (m), 3 females (f)), 2 weeks (turquoise) (*n* = 12, 7 m, 5 f), 3 weeks (green) (*n* = 18; 8 m, 10 f), 4 weeks (dark blue) (*n* = 14; 7 m, 7 f), control/CAR mice after TD-immunization. **c**–**e** Schematics of the general forms of the branched, linear, and null models of CAR⁺ MZB cell generation upon TD-antigen-mediated B cell activation. In-depth details about model specifics are given in the Materials and Methods. Source data are provided as a Source Data file (see sheets modeling data).

to calculate the NP-titers in sera of immunized control/CAR and N2IC/hCD2 animals using Excel 2010. All NP-specific titers are given as log10 relative units.

**ELISpot**

For antigen-specific ELISpot with splenic or BM cells, 96-well ELISpot plates (Millipore) were coated overnight at 4 °C with NP3– or NP14–BSA (Biosearch Technologies), diluted in carbonate buffer.

Plates were washed with PBS and blocked for 3 h at 37 °C with 10% FCS/ B cell medium (BCM). 5 × 10⁵ cells were seeded per well and incubated for 24 h at 37 °C. Plates were washed with PBS-T (PBS with 0.025% Tween20). Biotinylated antibodies against the detecting isotype were diluted in 1% BSA/PBS and incubated for 2 h at 37 °C. After washing with PBS-T, streptavidin horseradish peroxidase Avidin D (Vector) in 1% BSA/PBS was added and incubated for 45 min at room temperature. For the development of spots, a developing buffer was prepared as

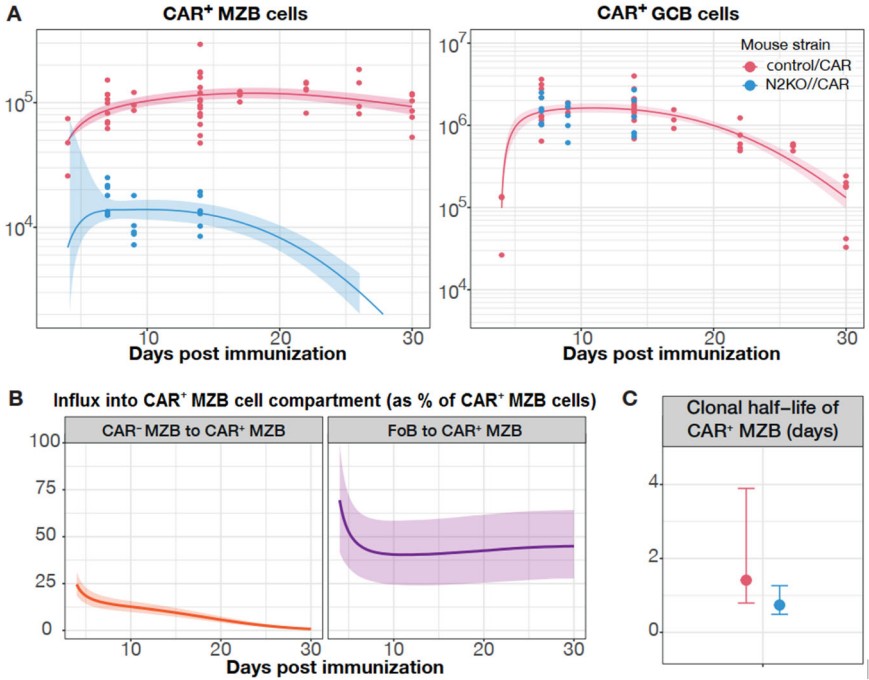

**Fig. 9 | Dynamics of CAR+ Marginal Zone B cells and Germinal Center B cells during a TD-immune response. a** Time-course of counts of CAR+ Marginal Zone B (MZB) and CAR+ Germinal Center B (GCB) cells post-immunization in control/CAR (red dots, $n = 53$) and N2KO//CAR (blue dots, $n = 18$) mice. We show the mean of the predictions from the best-fitting branched model with time-dependent antigen-activation of Follicular B (FoB) and MZB cells as smooth lines. The uncertainty in estimates of the mean is shown as shaded envelopes depicting 95% credible intervals (-2 SEM). The y-axis represents total cell counts. The sequential gating strategy to retrieve the frequencies of CAR+ GCB and CAR+ MZB cells is depicted in Supplementary Fig. 11. Control/CAR mice: d4 ($n = 3$; 2 males (m), 1 female (f)), d7 ($n = 9$, 5 m, 4 f), d9 ($n = 3$; 2 m, 1 f), d14 ($n = 15$; 8 m, 7 f), d17 ($n = 3$; 3 f), d22 ($n = 5$; 1 m, 4 f), d26 ($n = 4$; 2 m, 2 f), d30 ($n = 6$; 4 m, 2 f); N2KO//CAR mice: d7 ($n = 7$; 2 m, 5 f), d9 ($n = 5$; 1 m, 4 f), (d14 $n = 6$; 5 m, 1 f). **b** Estimate of the total daily influx into the CAR-expressing MZB cell subset, as percent of total CAR+ MZB cell pool sizes (y-axis). We show the mean values as smooth lines and the prediction error around mean is

shown as 95% credible intervals (shaded envelopes). The prediction error in the rate of influx from CAR- MZB to CAR+ MZB (left panel) is calculated across all time points using mice from both the control and N2KO groups, i.e., $n = 71$ mice. The uncertainty in the prediction of the mean value of the rate of influx from FoB to CAR+ MZB (right panel) is calculated using only control/CAR data ($n = 53$ mice). **c** Clonal half-life of CAR-expressing MZB cells in control/CAR (red) and N2KO//CAR (blue) mice. Clonal half-life is defined as the ln(2)/net loss rate ($\lambda$ for control/CAR group and $\lambda'$ for N2KO//CAR group). This quantity depicts the total time taken for the size of a clonal lineage to decrease by 2-fold. The uncertainty in the estimates of clonal half-life derives from uncertainty in estimating loss rates $\lambda$ across $n = 53$ control/CAR mice (red) and $\lambda'$ across $n = 18$ N2KO//CAR mice (blue). Plot shows the mean (circles) estimate with ±2 SEM (error bars) of the clonal half-life of CAR+ MZB cells. For both (**b**) and (**c**), time point-wise mouse specifics are the same as described in (**a**). Source data are provided as a Source Data file (see sheets modeling data).

follows: one tablet of each 3,3′- Diaminobenzidin peroxidase-substrate (0.7 mg/ml, Sigma-Aldrich, gold and silver tablets) was dissolved in 5 ml distilled water, the solution was pooled and filtered. 50 μl/well of this buffer was incubated for 8 min (splenocytes) or 12 min (BM cells). The reaction was stopped by washing with distilled water. Spots were visualized and counted with the ImmunoSpot Series 5 UV Analyzer (equipped with software version 5.0, CTL Europe).

### Flow cytometry
Single cell suspensions were prepared from the spleen and BM. Surface staining of lymphocytes with the corresponding antibodies in MACS buffer (Miltenyi) was performed on ice for 25 min. To omit dead cells from the analysis, cells were stained for 20 min on ice with LIVE/DEAD Fixable Blue Dead Cell Stain Kit (Invitrogen) prior to surface antibody staining. For intracellular FACS staining, cells were fixed with 2% par-aformaldehyde (1:2 PBS-diluted Histofix, Carl Roth) for 10 min at room temperature and permeabilized in ice-cold 100% methanol for 10 min on ice. Cells were incubated for 1 h at room temperature with the corresponding antibodies. Dead cells were excluded by staining for 5 min on ice with LIVE/DEAD Fixable Blue Dead Cell Stain Kit (Invitrogen) before fixation. Cytometry analysis was performed on a LSRII FACS Fortessa (BD Biosciences) coupled to the BD FACS DIVA Software V8.0.1. Results were evaluated using FlowJo (v9 and v10). Representative flow cytometry gating strategies for surface and intracellular

cell stainings to distinguish B lymphocytes and their subpopulations, reporter+ and reporter- cells, and plasmablasts and plasma cells among total live lymphocytes, can be found in Supplementary Fig. 14 a–e.

### Statistics & reproducibility
Model ranking and correlation analysis of CAR expression dynamics between FoB, MZB and GCB cells was performed in R. All other statistical analyses, including testing for distributions and equal variance, calculations of means and standard deviation (SD), determining $p$-values of 1-way or 2-way ANOVA with multiple comparisons tests and paired value tests were done with GraphPad Prism (version 9.5.1). $P$-values of less than 0.05 were considered significant. *$P < 0.05$, **$P < 0.01$, ***$P < 0.001$, and ****$P < 0.0001$. Error bars in all figures define means ± SD. Sample sizes were determined based on previous experiments and preliminary data. No statistical methods were used to pre-determine samples sizes. Our goal was to analyze at least 6 animals per group. In cases in which differences were pronounced enough to achieve significance with smaller sample sizes, adjustments were made accordingly. In general, we analyzed a minimum of 4 animals per genotype and treatment. Only in a few cases we analyzed only three animals. Most of the experiments with animal numbers ≤3 were performed during the revision phase, where we had only a limited number of mice available and were therefore unable to perform all the experiments requested by the reviewers with higher numbers of mice. Despite the

**Table 1 | a Comparison of models describing population dynamics of Marginal Zone B and Germinal Center B cell populations upon TD-immunization. b Parameter estimates from the best-fitting branched pathway with time-varying influx model**

| A | | | |
|---|---|---|---|
| | **Model weights (%)** | | |
| | **Time-varying influx** $\alpha, \beta$ = time-varying $\delta, \lambda, \mu$ = constant | **Neutral** $\alpha, \beta, \mu, \delta,$ $\lambda$ = constant | **Time-varying loss** $\delta$ = time-varying $\alpha, \beta, \mu, \lambda$ = constant |
| **Branched** | **79** | 0 | 0 |
| Linear | 7 | 0 | 0 |
| Null | 14 | 0 | 0 |
| **B** | | | |
| **Parameter** | **Estimates** | | |
| | **Median** | **95% Credible interval** | |
| Rate of influx of FoB into MZB (day$^{-1}$) | 1/1430 | (1/5500, 1/770) | |
| Time taken for the influx rate to halve (days) | 14.2 | (12.9, 15.4) | |
| Rate of influx of CAR$^-$ to CAR$^+$ MZB (day$^{-1}$) | 1/150 | (1/270, 1/100) | |
| Rate of net loss of MZB cells in Control/CAR (day$^{-1}$) | 0.49 | (0.18, 0.87) | |
| Rate of net loss of MZB cells in N2KO//CAR (day$^{-1}$) | 0.94 | (0.55, 1.4) | |

Relative statistical support for each model is indicated as % model weights (see Supplementary Note 1 for details on model weights estimation). Model-specific details are described in Fig. 8c–e. Here, $\alpha$ = rate of influx of Follicular B (FoB) cells into the Germinal Center B cell (GCB) compartment, $\beta$ = rate of upregulation of CAR expression on Marginal Zone B (MZB) cells, $\mu$ = rate of differentiation of FoB into CAR$^+$ MZB cells, $\delta$ = loss rate of CAR$^+$ GCB cells, $\lambda$ = loss rate of MZB cells. Best-fit model has the highest model weight (highlighted in blue bold font).

smaller number of mice, our analysis still yielded significant results or, at least, clear trends, as the biological differences between the samples were sufficiently robust. Sample sizes with only two animals were excluded from statistical analyses. Control and analytical mice were always analyzed in parallel and predominantly used in an age-matched manner, when possible. The experiments were repeated with different individual animals in at least two independent immunization or cell preparation rounds. No data were excluded from the analyses. The experiments were not randomized. The investigators were not blinded to allocations during experiments and outcome assessment. Sample sizes and the chosen statistical tests are indicated in each figure legend.

## Mathematical models of B cell dynamics during a TD-immune response

To model the emergence of CAR$^+$ MZB and GCB cells post-immunization in control/CAR and N2KO//CAR mice, we developed an array of system of ordinary differential equations (ODE) that explored diverse pathways of B cell differentiation and logistics of their maintenance post activation. We observe no significant differences in the counts of CAR$^+$ MZB and GCB cells up to day 4 post-immunization and therefore anchor the initial condition $t_0$ = 4 days in all our models. Model schematics are described in Fig. 8.

## Mechanisms of maintenance of GCB cells

We assumed that all GCB cells are derived from antigen-activated FoB cells with a *per capita* rate $\alpha$. We explored modeling possibilities in which either the influx of cells into the GC compartment or their loss from the GC pool are varying with time post-immunization.

**Neutral dynamics.** This is the simplest model, which assumes constant birth-loss. The rates of influx into the GC ($\alpha$) and net loss of cells from the GC ($\delta$), remain constant with time. In all our models, the net loss rate $\delta$ is the balance between production of new GCB cells by cell division and their 'true loss' by death and differentiation.

**Time-dependent loss of GCB cells.** This model assumes that the net loss rate of GCB cells varies with time post-immunization. We explored the sigmoid form of $\delta(t)$, as shown in Eq. 1 below. The influx of antigen-activated FoB cells into the GC compartment $\alpha$ is assumed to be

constant in this model. Parameters $\nu 1$ and $\delta_0$ are unknowns in this model and are estimated from model fits, along with $\alpha$.

$$\delta(t) = \frac{\delta_0}{1 + e^{\nu 1(t-t_0)^2}} \qquad (1)$$

**Time-dependent recruitment of FoB cells into GC.** In this model, the *per capita* rate of recruitment of FoB cells into the GC varies with time $\alpha(t)$ and is defined as,

$$\alpha(t) = \frac{\alpha_0}{1 + e^{\nu 2(t-t_0)^2}} \qquad (2)$$

The net loss rate of GCB cells $\delta$ is constant in this model, estimated from the model fits along with $\nu 2$ and $\alpha_0$.

## Loss of CAR$^+$ MZB cells

We assumed a constant *per capita* rate of net loss $\lambda$ for CAR$^+$ MZB cells. It has been shown that persistent Notch2 signaling is required for B cells to maintain their MZ status[31,32]. Therefore, the net loss rate of MZB cells reflects the balance between their self-renewal (division) and their true loss via death, differentiation and deprivation of Notch2-dependent signals. In N2KO//CAR mice, antigen-activation of B cells leads to deletion of Notch2 and thus increases the propensity of their loss by deprivation of Notch2-mediated signals. Therefore, we assume different loss rates of CAR$^+$ MZB cells in control/CAR mice and N2KO//CAR mice *viz.* $\lambda$ and $\lambda'$.

## Mechanisms of generation of MZB cells

We explored three different models − branched, linear, and null − to explain the emergence of CAR$^+$ MZB cells after immunization.

**Branched model.** In this model, the antigen-dependent activation of FoB cells induces a branch point in the B cell differentiation pathway, where some cells participate in GC reactions, while others acquire a MZB cell phenotype (Fig. 8c). Since the dynamics of CAR expression on FoB cells are unclear, we considered that all antigen-activated FoB (F) cells (irrespective of their CAR expression status) differentiate into CAR$^+$ MZB cells (M) with a *per capita* rate $\mu$. The dynamics for CAR$^+$ GCB (G) cells are identical between control/CAR and N2KO//CAR mice.

The general form ODE system depicting the branched pathway is defined as,

$$\dot{G} = \alpha(t)F(t) - \delta(t)G(t)$$

$$\dot{M} = \mu(t)F(t) + \beta(t) - \lambda(t)M(t)$$

$$\dot{M}_{N2} = \beta(t) - \lambda'(t)M_{N2}(t) \tag{3}$$

where $M_{N2}$ represents CAR$^+$ MZB cells in N2KO//CAR mice.

**Linear model.** This model follows a linear path in which antigen-activated FoB cells differentiate into GCB cells, which then subsequently give rise to CAR-expressing MZB cells (Fig. 8d). The general form of ODE system of the linear pathway is given as,

$$\dot{G} = \alpha(t)F(t) - \delta(t)G(t)$$

$$\dot{M} = \mu(t)G(t) + \beta(t) - \lambda(t)M(t)$$

$$\dot{M}_{N2} = \beta(t) - \lambda'(t)M_{N2}(t) \tag{4}$$

**Null model.** This model assumes that the CAR$^+$ fraction in MZB cells is purely derived from activation of pre-existing MZB cells and is maintained by activation-induced continuous new deletion of the stop cassette in CAR$^-$ MZB cells and by self-renewal of newly generated CAR$^+$ MZB cells (Fig. 8e).

$$\dot{G} = \alpha(t)F(t) - \delta(t)G(t)$$

$$\dot{M} = \beta(t) - \lambda(t)M(t)$$

$$\dot{M}_{N2} = \beta(t) - \lambda'(t)M_{N2}(t) \tag{5}$$

We assumed that the rate of activation of CAR$^-$ MZB to CAR$^+$ MZB cells ($\beta$) varies with time as shown below.

$$\beta(t) = \frac{\beta_0}{1 + e^{\nu 3(t - t_0)^2}} \tag{6}$$

Lastly, we explored the possibility that the rate of influx of antigen-activated FoB cells to CAR$^+$ MZB cells (defined as $\mu$) may vary with time, likely due to the dependence on antigen availability in the cellular environment. We used the same form as shown in Eq. 6 to describe the time dependence in $\mu(t)$.

### Reporting summary
Further information on research design is available in the Nature Portfolio Reporting Summary linked to this article.

## Data availability
All source data are provided with this paper as a Source Data file. Previously generated RNA sequencing/microarray data, which were analyzed in this study, have been deposited in ENA under the accession code PRJEB35207 (Supplementary Fig. 1c) or at Zenodo under the https://doi.org/10.5281/zenodo.10491127 (https://zenodo.org/records/10491127) (Fig. 1c)[61]. Graphs from Supplementary Fig. 8b were produced with data derived from the ImmGen.org databrowser "Gene Skyline − RNAseq" (http://rstats.immgen.org/Skyline/skyline.html)[62]. Source data are provided with this paper.

## Code availability
Code defining all our models and scripts defining our plotting and statistical routines are available at a Github repository integrated in the Zenodo records (https://zenodo.org/records/10475721)[63]. For high reproducibility, we provide definitions of the prior distributions of the unknown variables and the descriptor functions used to evaluate fixed variables, in our model code.

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

## Acknowledgements

This work was supported by the Deutsche Forschungsgemeinschaft (DFG ZI1382/4-1), U.Z.-S. and the National Institute of Allergy and Infectious Diseases of the National Institutes of Health under award number RO1AI170965, S.R. We thank the animal facility of the Helmholtz Center and our animal caretaker team for the excellent housing of the mice and Krisztina Zeller for excellent technical help with murine DNA preparation and genotyping. This work benefitted from data assembled by the ImmGen consortium.

## Author contributions

T.B. performed most experiments and analyzed data. M.L., S.E., UR performed experiments. A.J.Y. contributed to mathematical modeling and manuscript editing. M.S.-S. contributed to data interpretation and provided material. S.R. developed and performed mathematical modeling. S.R., L.J.S. and U.Z.-S. designed research and contributed to data interpretation and manuscript preparation. T.B. and U.Z.-S. wrote the manuscript. L.J.S. and T.B. prepared the figures from the experimental part and calculated the statistics. S.R. wrote the mathematical modeling part and prepared the figures. S.R., L.J.S. and U.Z.-S. contributed equally to supervising the study (L.J.S. and U.Z-S.: experimental part; S.R. mathematical modeling part). All authors read and approved the final paper.

## Funding

## Competing interests

The authors declare no competing interests.
