## [Peer Review File · Nature Communications]

Notch2 controls developmental fate choices between germinal center and marginal zone B cells upon immunizationREVIEWER COMMENTS

Reviewer #1 (Remarks to the Author):

In this manuscript Babushku et al. characterize the role of Notch2 in differentiation of mature B lymphocytes into GC-, MZ- and plasma B cells. Notch reporter mice are combined with gain- or loss-of-function mice of the Notch2 receptor, which plays a key role in this process. Clearly, GCB cell differentiation depends on presence and activation of Notch2, as convincingly and elegantly shown in this study. Importantly, the Notch2 expression is upregulated upon activation at the right stage and IRF4 expression is induced. T-cell dependence is also analyzed.

While technically the majority of experiments are convincing, the novelty in comparison to their already published data in 2021 (Lechner et al.) is less apparent.

Major points:

1) Figure 1, regarding the Notch-reporter (CBF:H2B-Venus): Histones are very stable proteins. Do you know about the half-life of H2B-Venus? Are several cell divisions needed to dilute the signal? Otherwise such a broad upregulation is to be expected.

2) Regarding Fig-3 and Fig-4: The constitutive expression of NICD2 is somewhat artificial. Do you have any additional evidence for endogenous Notch2 activation? In fact, is the cleavage of the Notch2 receptor required and, if yes, what is the evidence? Are 'classical' Notch target genes expressed?

3) IRF4 induction page 12 / 13: Mechanistically the statement that Notch2 signaling induces IRF4 expression is important. Do you have any evidence whether IRF4 is a direct Notch target genes?

4) Ligand for Notch2: It remains unclear whether the ligand for Notch2 is Delta-1 or Jag1. This aspect should be at least discussed. Maybe the authors' data would also implicate more the one or the other.

Overall evaluation:

Together, this manuscript confirms the role of Notch2 in the follicular-B and marginal zone B-cell decision. My major criticisms are that the novelty is fairly limited (compared to Lechner et al. 2021) and the major points rely on NICD2 overexpression.

Reviewer #2 (Remarks to the Author):

In the manuscript entitled, "Notch2 signaling guides B cells away from germinal centers towards marginal zone B cell and plasma cell differentiation," Babushku and colleagues perform experiments examining the frequency and intensity of B cell subpopulations undergoing Notch2 signaling in vivo using a Notch2 reporter mouse and also examine the consequences of loss of Notch2 or expression of constitutively active Notch2 in activated, GC, and post-GC B cells. The major findings from the manuscript are:

1. Notch signaling reporter expression is present in most follicular B cells and MZB and absent in most GC B cells
2. ~1/3 of MZB lose Notch2 reporter expression (although it is unclear what the functional consequences of this loss are, discussed further below).
3. Notch2 signaling in activated B cells can drive differentiation of cells into an MZ-like state during T-dependent immune responses.

The authors propose that Notch2 signals received in the B cell follicle can promote MZ B cell transdifferentiation from FoB or from activated B cells in T dependent immune responses and that Notch2 signals received in the follicle sustain MZ B cell persistence. Overall, the manuscript presents some potentially interesting findings but these findings are not explored with sufficient depth.

Major concerns:

1. What is the anatomic distribution of Venus negative MZB? There appears to be some discordance in the frequency of Venus positive MZB on FACS and IF section. On the FACS roughly 70% of MZB are high for Venus, whereas on the IF image (Fig. 1C), a minority of IgM+ cells in the MZ are negative for Venus expression. How do the authors account for this discrepancy. One possibility is that MZ B cells have increased Venus expression when they are inside the follicle and reduced expression in the MZ. This could be addressed via an in vivo labeling approach to compare Venus expression on blood exposed and non-blood exposed MZ B cells (PMID: 18037889). On the other hand, IF may only be sensitive enough to detect the highest expression of Venus, in which case the authors may wish to be more circumspect in their interpretation of the imaging.
2. What is the consequence of loss of Venus expression in ~1/3 of MZB? Do these cells lose their MZ phenotype? Are they enriched for cells that are more likely to undergo death? Are there differences in gene expression between Venus negative and Venus positive MZB.
3. Past studies have demonstrated that acute disruption of Notch2 signals leads to loss of MZB (PMID: 24323359). The authors propose that access to Notch2 ligands in the follicle may be important for sustaining the MZ state. If MZB cells lose Notch2 signaling in the anatomic MZ and Notch2 signaling when MZB enter the follicle is important for sustaining the MZ state, why do MZ B cells persist in the absence of CXCR5/Cxcl13 where MZB cannot access the follicle and are present only in the anatomic MZ (PMID: 18037889 and PMID: 15184895).
4. How dynamic is Venus expression in Notch2 signaling reporter mice? If ~20% of MZB cells exchange between the follicle and the anatomic MZ over the course of 1 hour, are Venus negative MZ B cells in the anatomic MZ truly an indicator of lack of ligand in this compartment. It would be extremely helpful to understand the dynamics of Notch-reporter expression in B cells in this model in order to know how to interpret the meaning of a reporter negative MZ B cell.
5. On FACS, Venus expression appears to be uniform in FoB (Venus expression is present in FoB but still at least 5 fold-lower than the highest MZB B cells), how does this uniform amount of reporter expression inform cell fate decisions of FoB transdifferentiating into MZB? A uniform amount of Notch-reporter expression suggests to me that while Notch2 may be required for transdifferentiation it is not sufficient to drive this process and that other signals are more important for guiding cell fate decisions. It would also be helpful for the authors to show MFI values for reporter expression in the various B cell subsets— compared to developing B cells in the bone marrow, reporter expression appears to be ~50-fold higher in 70% of MZ B cells.
6. Although a fraction of GC B cells in the LZ still express Venus reporter, there appears to be no functional consequence to loss of Notch2 signaling on GC dynamics itself. Are the authors suggesting that there is a physiologic function of Notch2 signaling in the LZ driven by FDC or Tfh?
7. In figure 3e the Cg1-cre reporter+CD95+CD38+ cells likely comprise a mixture of both recently activated cells and memory B cells rather than exclusively “activated non-GCB cells” the authors should include additional markers to more fully characterize this cell population or should modify their statement.
8. In Figure 4E-G, the authors conclusions seem to be somewhat of an over-interpretation of the data. The data support the conclusion that when Notch2 signals are not appropriately down-regulated as in the N2IC animals activated cells are arrested in the pre-plasmablast state but it does not necessarily follow that under physiologic circumstances strong Notch2 signaling in FoB promotes a pre-plasmablastic state. In order to support this statement the authors should examine pre-plasmablast/plasma cell responses in the N2KO//CAR mice. If pre-plasmablasts and/or plasma cell responses are not reduced in the N2KO mice then it would appear more likely that the data shown in 4E-G are the result of non-physiologic expression of the N2IC allele. Moreover, mechanistically, why do persistent Notch2 signals in the N2IC mice arrest these cells in a pre-plasmablast state?
9. Figures 5 and 6 offer a nice illustration that both enforced Notch2 signaling can drive MZ differentiation of activated B cells and that the generation of these cells is dependent on post-activation Notch2 signaling. The authors then use mathematical modeling to suggest that these T-dependent MZB cells arise from activated B but not GC B cells. I think the manuscript would benefit from a more in depth analysis of these cells. Can the authors provide additional experimental evidence to define the cellular ontogeny of these cells (I.e. that in WT mice they are not derived from GC B cells)? Are antigen-specific (NP-binding) cells present in T-dependent MZ B cells in similar numbers to MBC that are not present in the MZ? What is the contribution of these cells to a recall immune response (I.e. in a N2KO mouse is there a reduction in low affinity or high-affinity NP-specific antibody production at memory time point)?
10. It would be helpful for the authors to examine Notch-signaling Venus expression in activated B

cells as well as T-dependent MZ B cells. Do they show similar amounts of expression of the Venus reporter as resting FoB and MZB?

Minor concerns:

1. MZ B cells in the mouse are non-recirculating but are highly migratory . ~40% of MZ B cells can be visualized exchanging between the follicle and MZ within 1 hour (~20% move between the follicle and MZ and ~20% move between the MZ to the follicle) (PMID: 23263181). Please change the sentence in the introduction: "FoB cells recirculate between lymphoid organs, whereas MZB cells are mostly sessile in the marginal zone (MZ) and can occasionally migrate between the B cell follicle and the MZ (1)" to more accurately reflect the findings of the citation.
2. Data in Figure 3b and c using *cg1-cre/N2IC* mice is largely confirmatory of what has been previously shown by Valls et al (Supplementary Figure 3 in PMID: 28232365). Please indicate this in the text when discussing this data.
3. Please label x-axis in all histograms in Supply. Fig. 2b
4. Can the authors examine Notch2 MFI on LZ GCB versus DZ GCB? If CD40/BCR is driving Notch2 expression then it is more likely that Notch2 expression would be increased on LZ. It would also be helpful to see if there is a correlation between Notch2 MFI and Venus reporter expression in LZ and DZ GCB.
5. The data in figure 4A-C regarding mutual antagonism of Notch2 and Bcl6 has also been demonstrated by Valls et al. Please modify the sentence "These findings indicate that Notch2 signaling interferes with GC formation through Bcl6 suppression" to cite the Valls et al study (PMID: 28232365) appropriately.
6. When discussing antigen-dependent generation of MZB (page 29), please also cite PMID: 35768001 which has clearly demonstrated MZ-like memory B cells arising following viral infection in mice.

Reviewer #3 (Remarks to the Author):

The authors use a population model to augment their discussion of the various measurements of different B cell subsets as a function of time and genetic modification. In general their procedure is described adequately, the approach makes sense, and the final choice of model via assessing the accuracy of leave-one-out predictions is in line with standard treatments. Thus there were no fundamental problems here. I do however have some questions and comments which it would be worth responding to before publication.

- Are the " v " parameters which appear throughout (for example in supplemental equation 1 and main text equations 2 and 6 taken to be the same). If not, can they be assigned separate symbols to avoid confusion.
- it was unclear how the various processes in the model should depend on the existence of Notch2 signaling. If I correctly understood the details, aside from the decay of the CAR+MZB cells, these parameters seem to be taken to be the same for the "wild-type" and the knockout. It was not clear what the basis was for this assumption. why for example was the μ production rate not dependent on Notch2 signaling given the statement in the discussion that this signaling induces MZB development. In general, the model choices could have been connected better to the underlying biology.
- In general, I believe that mathematical models should try to predict new features rather than just recapitulate existing data. Are there any new predictions for further experiments that the authors can propose, as this would increase the usefulness of the modeling section.

Comments to reviewers #1–3

Dear reviewers,

Thank you very much for the evaluation of our manuscript. We greatly appreciate your comments and suggestions, which have significantly improved the manuscript. We have addressed all points comprehensively in our revised manuscript. Please, find our replies to your specific concerns and comments below.

Comments to reviewer #1

In this manuscript Babushku et al. characterize the role of Notch2 in differentiation of mature B lymphocytes into GC-, MZ- and plasma B cells. Notch reporter mice are combined with gain- or loss-of-function mice of the Notch2 receptor, which plays a key role in this process. Clearly, GCB cell differentiation depends on presence and activation of Notch2, as convincingly and elegantly shown in this study. Importantly, the Notch2 expression is upregulated upon activation at the right stage and IRF4 expression is induced. T-cell dependence is also analyzed.

While technically the majority of experiments are convincing, the novelty in comparison to their already published data in 2021 (Lechner et al.) is less apparent.

We thank reviewer #1 for the careful evaluation of our manuscript and his valuable comments and questions. We believe that we were able to adequately address all concerns raised by reviewer #1. In our revised manuscript, we also improved the emphasis on innovation and the advance from previous studies. The addition of new experiments suggested by the reviewers has further strengthened this aspect.

Below we outline the novel concepts described in our manuscript:

- (i) In our previous work (Lechner et al., 2021; PMID: 33597542) we describe the Notch-IC-mediated differentiation of FoB to MZB cells and showed in transplantation experiments that this process also occurs physiologically. However, the critical question of when FoB cells transdifferentiate into MZB cells during normal physiology remains open. In our present manuscript we show that transdifferentiation of FoB into MZB cells occurs during antigen-specific B cell immune responses. Our mathematical modelling suggests that Notch signaling triggers a binary switch in antigen-activated B cells: B cells that downmodulate or lose their Notch signal develop into GCB cells; in contrast, activated B cells with sustained and enhanced Notch2 signaling develop into MZB cells and/or immature plasma cells. **This point is now highlighted in the first chapter of the discussion.** Our data further showed that upon TD-immunization Notch2-expression and signaling is upregulated in activated B cells, as additionally evidenced by the analysis of the Notch-target gene Hes1 (**new Fig. 4c**). This suggests that the interplay between BCR, CD40 and Notch signaling determines this binary cell fate decision.

- (ii) By analyzing H2B-Venus expression in Notch-reporter mice we provided the first evidence for active basal Notch signaling in all immature and mature B2 cells, except for GCB cells and plasma cells. To strengthen this point beyond the reporter transgene, we added new data in the revised manuscript demonstrating that Notch-target genes are more strongly expressed in FoB cells from WT mice compared to FoB cells from Notch2-deficient mice (**new Fig. 1c**).
- (iii) We found that Venus-expression and Hes1-expression are already elevated in pre-MZB cells, compared to FoB cells, and are highest in mature MZB cells (**new Fig. 1d**). This underscores our hypothesis that only cells which receive strong Notch2 signals develop into MZB cells and is in line with our observation that Notch-target genes are higher expressed in MZB in comparison to FoB cells (**new Supplementary Fig. 1c**).
- (iv) Finally, we show for the first time that Notch signaling is dispensable for the initiation and maintenance of the GC reaction (Fig. 3c), however seems to be reactivated during positive selection of GC B cells, contributing to the expansion of IgG1⁺ B cells (**new Fig. 3f and g**).

Major points of Reviewer #1

Point 1: Figure 1, regarding the Notch-reporter (CBF:H2B-Venus): Histones are very stable proteins. Do you know about the half-life of H2B-Venus? Are several cell divisions needed to dilute the signal? Otherwise such a broad upregulation is to be expected.

We agree with the reviewer that fusion proteins with histones are generally very stable. However, recent publications discussed and presented data, which show degradation of H2B and other core histones as a regulatory tool at certain cellular differentiation or activation states (Qian XM et al., 2013; Liu Y et al., 2021; Chen YS and Qui XB, 2012; PMIDs: 23706739, 33796843, 23177144). Such processes may result in destabilization of the H2B-Venus fusion protein.

We contacted the authors of the original manuscript, describing the Notch-reporter mice (Nowotschin et al., 2013; PMID: 23617465), but they did not have any information about the stability of the H2B-Venus fusion protein in lymphocytes.

[REDACTED]

To monitor the stability of H2B-Venus in B cells, we isolated splenic B cells and stimulated them ex vivo with LPS for 3 days to induce proliferation and plasmacytic differentiation, as H2B-Venus-expression is strongly downregulated in plasma cells (PC). Prior to LPS treatment, the cells were stained with Cell Trace Far Red to determine extent of their proliferation. We found that (i) H2B-Venus was stable in non-dividing or slowly dividing B cells for at least 72 hours. (ii) In activated and proliferating B cells the H2B-Venus protein was reduced around 10-fold in non-plasmablasts and almost completely in in-vitro differentiated plasmablasts (PB). The decrease in Venus expression did not (PB/PC) or only partially (all B220⁺ B cells) correlate with the extent of cell division. This argues against an exclusive dilution-dependent H2B-Venus reduction by cell division, but rather suggest that H2B-Venus becomes unstable under certain conditions, for example upon B cell activation.

Editorial Note: Some text above has been redacted to remove reference to personal communication where no permission to publish could be obtained.

Supplementary Fig. 7b: Venus-expression is downregulated upon LPS stimulation. Ex vivo isolated B cells were stimulated with LPS for 3 days. Prior to stimulation, they were stained with Cell Trace Far Red to determine the extent of their proliferation. To determine the Venus expression dependent on proliferation the Far Red staining was subdivided in 5 fractions as indicated in the left histogram overlay. In each fraction the Venus expression was determined in PB and non-PB (B cells), which were gated as indicated in the zebra plots. The histogram overlays show the Venus expression in PB in fractions 2-4 and non-PB in fractions 0-4.

From these data we conclude that the H2B-Venus fusion-protein is in principle highly stable in developing B cells and FoB cells. We also agree with the reviewer that this may lead to accumulation of H2B-Venus if B cells receive a recurrent Notch signal or if Notch-IC remains in the nucleus for an extended period of time.

Different mechanisms may be responsible for the loss of H2B-Venus expression in individual cell types and/or differentiation stages. (i) In GCB cells, it could be due to rapid proliferation combined with active silencing of Notch-signaling. (ii) Our new data suggest that the loss of H2B-Venus in developing plasma cells is a combined effect of H2B-Venus degradation in activated B cells primed towards PC differentiation and downregulation of Notch signaling during terminal maturation stages (**new Supplementary Fig. 7b**). (iii) In some MZB cells, H2B-Venus appears to be highly unstable. The reason may be that H2B, due to regulatory mechanisms of histones in pre-activated (e.g., MZB cells) and activated B cells (pre-plasmablasts), is destabilized (Qian XM et al., 2013; Liu Y et al., 2021; Chen YS and Qui XB, 2012; PMIDs: 23706739, 33796843, 23177144). Our data shown in the **new Fig. 2a and c** suggest that MZB cells have a higher Notch signaling than FoB cells as indicated by higher Hes1 levels, but H2B-Venus appears to be lost in some MZB cells, possibly through histone regulatory mechanisms, cell division, or onset of PC differentiation in a certain proportion of MZB cells.

The stability of H2B-Venus is discussed in the second paragraph of the discussion in the revised manuscript.

Point 2: Regarding Fig-3 and Fig-4: The constitutive expression of NICD2 is somewhat artificial. Do you have any additional evidence for endogenous Notch2 activation? In fact, is the cleavage of the Notch2 receptor required and, if yes, what is the evidence? Are 'classical' Notch target genes expressed?

Figure 3: Notch2 signaling in GC:

In the previous Figure 3, we showed that Notch2 signaling is dispensable for GC development (N2KO//CAR) and that transgenic activation of Notch blocks induction of GCs (N2IC/hCD2). Our experimental and mathematical data suggest that Notch2 signaling is upregulated in antigen-activated B cells, where it acts as binary switch: Activated B cells that counter-regulate and silence Notch signaling develop to GCB cells, while activated B cells with sustained Notch2 signaling develop to MZB cells and PC, dependent on their intracellular Irf4-level.

To confirm our assumption, we studied the expression of Hes1, a classical Notch-target gene, in antigen-activated B cells from N2IC, N2KO and control mice, as suggested by the reviewer. We found that Hes1 expression follows a similar kinetics as Notch2 surface expression (Figure 4b-c). Thus, in wildtype mice, Hes1 was significantly higher expressed in activated B cells (CD38⁺CD95⁺) compared to non-activated CD38⁺CD95^{low} and GC B cells. No upregulation of Hes1 in activated B cells was detected in N2KO mice, confirming a critical role for Notch2 in this process. In N2IC mice, Hes1 regulation was similar as in controls but generally higher, suggesting combined endogenous and transgenic Notch signaling (**new Fig. 4c and supplementary Fig. 5d**). Overall, these results confirmed that Notch2 signaling is enhanced in activated B cells and counter-regulated in GCB cells.

Fig. 4b and c: Increased expression of Notch2 and Hes-1 upon TD immune responses in activated B cells. Notch2 and Hes1 expression was determined by flow cytometry. **(b)** Fold induction of Notch2 cell surface expression (MFI) expression in the indicated populations of CAR⁺ cells from control/CAR mice, relative to its expression levels in CAR⁻ B cells (MFI set to 1, depicted by the dotted line). **(c)** Fold induction of Hes1 (MFI) in the indicated populations of Cre-reporter⁺ cells from the indicated genotypes, relative to its expression levels in Cre-reporter⁻ B cells (MFI set to 1, depicted by the dotted line)

Figure 4: Notch-signaling in plasma cell differentiation:

In the previous Figure 4e-g we showed that N2IC/hCD2 mice have increased amounts of plasmablastic cells. We concluded from these data that N2IC drives PC differentiation. Since we based our assumption on the study of enforced transgenic Notch2IC expression, we agree with the reviewer that our data should be validated. In the revised manuscript we additionally analyzed plasma cell differentiation in GCB cell-

specific Notch2 knockout (N2KO//CAR// γ 1-Cre) mice and in B cell-specific Notch2 knockout (N2KO//CD19-Cre) mice. We found that 7 days after TD-immunization, percentages of PC were reduced in both strains of Notch2-deficient mice (see also point 8 from reviewer 2), whereas no difference was observed at later time points (**new supplementary Fig. 6b-e**). These data suggest that Notch2 signaling appears to have an effect mainly on the early wave of PC differentiation. Both reviewer #1 and reviewer #2 suggested that we should have a closer look at the role of Notch2 signaling during PC-differentiation. The addition of novel data derived from experiments proposed by reviewers #1 and #2, suggests that Notch2 signaling drives the expansion of pre-plasmablasts, but has to be switched off to allow for terminal PC differentiation (outlined in detail in the response to reviewer #2 point 8). To make this point clearer, we reorganized our figures and compiled all data pertaining to this hypothesis in **Fig. 5 and supplementary Fig. 6 and 7**.

Evidence that cleavage of the Notch-receptor is necessary for plasma cell differentiation can be found in the literature, where GCB cells have been shown to be impaired in their plasmablastic differentiation in the presence of the γ -secretase inhibitor DAPT in an in vitro culture system which promotes plasmablast differentiation (Yoon et al., 2009; PMID: 19542446).

Point 3: IRF4 induction page 12 / 13: Mechanistically the statement that Notch2 signaling induces IRF4 expression is important. Do you have any evidence whether IRF4 is a direct Notch target gene?

We detected a slight upregulation (about twofold) of *Irf4* mRNA upon Notch2IC induction in quiescent FoB cells (Lechner et al., 2021; PMID: 33597542), indicating that strong Notch signaling results in upregulation of *Irf4*. However, we do not have any CHIP-seq or other experimental data, which would prove an interaction of Notch2IC at the *Irf4* promoter.

We have included this point in the discussion in the revised manuscript.

Point 4: Ligand for Notch2: It remains unclear whether the ligand for Notch2 is Delta-1 or Jag1. This aspect should be at least discussed. Maybe the authors' data would also implicate more the one or the other.

Fasnacht et al. suggested that the interaction between Dll-1 and Notch2 determines the development of MZB cells. This was demonstrated in conditional Dll-1KO mice in combination with CCL-19-Cre (which leads to the ablation of Dll-1 in follicular dendritic cells, MAdCAM+ marginal reticular cells, and B cell zone reticular cells (Fasnacht et al., 2013; PMID: 25311507)). **We included this information in the introduction of the revised manuscript.**

It is less clear which receptor/ligand interaction is necessary for the maintenance of MZB cells. Application of an antagonistic Notch2 antibody leads to the loss of MZB cells with a half-life of 24-36 hours underlining the importance of Notch2 signaling in the maintenance of MZB cells as well. In the murine spleen, Dll-1 expression has been detected only inside the follicle but not in the MZ (Fasnacht et al., 2013; PMID: 25311507), implying that MZB cells have to migrate into the follicle to refresh their Notch2 signal by interacting with Dll-1-expressing cells. Alternatively, ligands other than Dll-1 could be responsible for the maintenance of MZB cells through Notch2 signals. For example, Dll-4 has been shown to be strongly expressed by endothelial cells of the marginal sinus (Tan et al. 2009; PMID: 19217325). To determine where MZB cells receive their Notch2-dependent maintenance signal is an important point, which has to be further evaluated in a follow-up study in the future. **We included these aspects in the discussion of the revised manuscript.**

Up to date, a defect in plasma cell differentiation has been mainly described for Notch1-deficient mice, which show an impaired immune response after immunization with NP-LPS and NP-Ficolin (Kang et al., 2014; Zhu et al., 2017; PMIDs: 24913005, 28863329). Notch1-deficient B cells are also impaired in their PC differentiation in vitro in the presence of Dll-1 expressing cells. PC differentiation also appears to be mediated mainly by Dll-1, since co-stimulation with Dll-1 expressing cells enhanced PC differentiation, whereas co-stimulation with Jagged-1 diminished PC differentiation (Santos et al., 2008; PMID: 17878313). **We have included this point in the discussion of the revised manuscript.**

Comments to reviewer #2

We thank reviewer's #2 for his constructive criticism and the helpful suggestions to improve the overall quality of our manuscript. We addressed his points as detailed below.

Major concerns:

1. What is the anatomic distribution of Venus negative MZB? There appears to be some discordance in the frequency of Venus positive MZB on FACS and IF section. On the FACS roughly 70% of MZB are high for Venus, whereas on the IF image (Fig. 1C), a minority of IgM+ cells in the MZ are negative for Venus expression. How do the authors account for this discrepancy? One possibility is that MZ B cells have increased Venus expression when they are inside the follicle and reduced expression in the MZ. This could be addressed via an in vivo labeling approach to compare Venus expression on blood exposed and non-blood exposed MZ B cells (PMID: 18037889). On the other hand, IF may only be sensitive enough to detect the highest expression of Venus, in which case the authors may wish to be more circumspect in their interpretation of the imaging.

Distribution of H2B-Venus-high and H2B-Venus-low MZB cells

To address the reviewer's concerns, we re-analyzed IF sections and now show in one image only the Venus-expression without overlaying the IgM staining, as well as a magnification of a specific area of the section (**Figure 2b**). These images demonstrate that both Venus^{high} and Venus^{low} are found within the follicle as well as in the MZ. At both locations, some cells do not display detectable H2B-Venus expression, so that they appeared to be Venus^{neg} (yellow arrows). Some of the H2B-Venus^{high} cells in the follicle seemed to have a MZB cell shape (bigger in size and less roundish, (Arnon et al., 2013; PMID: 23263181) (white arrow). These may be newly generated MZB cells or MZB cells that migrated into the follicle and received a Notch2 signal there. Our data may indicate that both Venus^{high} and Venus^{low} cells migrate between the MZ and follicle and stochastically receive a Notch signal inside the follicle, subsequently becoming Venus^{high} again.

Fig. 2b: Some MZB cells downregulate the Venus signal. Immunofluorescence analysis of splenic sections stained for IgM⁺ (red), and metallophilic macrophages MOMA1⁺ (blue). Green dots represent Venus-expressing cells. In the lower part an amplification of the sections is shown. Venus^{low} cells are marked with a yellow arrow and Venus^{high} cells with a white arrow.

In vivo labelling approach to compare Venus expression on blood exposed and non-blood exposed MZ B cells

Instead of the reviewer's suggestion to label all cells outside the follicle by injecting CD21-PE, we injected the mice with FTY-720 in order to relocate all cells into the follicle. Our hypothesis was that all MZB cells would then become Venus^{high}.

48 hours after FTY-720 injection, all B cells were located inside the follicle (IHC stainings, data not shown). We observed that in the presence of FTY-720 MZB cells still separated into Venus^{high} and Venus^{low} MZB cells with a slight increase of the Venus^{high} fraction. Interestingly, FTY-treatment resulted in a clear upregulation of Hes1 in FoB and pre-MZB cells, while H2B-Venus expression in these populations was unchanged. The reason could be that H2B-Venus upregulation is slower than Hes1 upregulation and thus runs behind the changes in Hes1 expression. This could also explain why Hes1 levels are already equal in pre-MZ and MZ cells, while Venus expression is still increasing (Fig. 1d) from pre-MZB to MZB cells. We conclude from these data that the retention of B cells in the follicle increases the likelihood of a Dll-1/Notch2 interaction of FoB cells. Both the Hes1 and H2B-Venus-expression do not really change in MZB cells when they are located in the follicle, suggesting that MZB cells already have a strong Notch2 signal which is not further enhanced by translocating them to/retaining them in the follicle. Translocation of MZB cells into the follicle appears not to revert H2B-Venus instability in a part of MZB cells.

Information for the reviewer: Treatment of CBF:H2B-Venus mice with FTY-720 upregulates Hes1 in pre-MZB and FoB cells, but does not prevent some MZB cells from being Venus^{low}. Mice were treated with FTY720. Splenic cells were prepared after 48h. Venus and Hes1 expression in FoB, pre-MZB and MZB cells (gated as shown in the contour plots) are shown in the histogram overlays in the absence (0 h control) and presence of FTY-720 treatment. The histogram overlays show the Venus and Hes1 expression in the indicated B cell population with and without FTY-720 treatment. N=3 untreated and 2 treated mice

This information is only for the reviewer.

We have not included these data in the revised manuscript, because, as they do not answer the question whether Venus^{high} cells are mainly located in the follicle and why some MZB cells lose their Venus signal. However, we wanted to share these findings with the reviewer because they strengthen our original assumption that FoB cells receive their Notch signal in the follicle and subsequently develop into MZB cells. In the future, it will be interesting to identify the spatial location at which MZB cells receive their maintenance signal.

2. What is the consequence of loss of Venus expression in ~1/3 of MZB? Do these cells lose their MZ phenotype? Are they enriched for cells that are more likely to undergo death? Are there differences in gene expression between Venus negative and Venus positive MZB cells?

According to the reviewer's suggestion, we determined the phenotype of Venus^{high} and Venus^{low} MZB cells in more detail. In comparison to Venus^{high} cells, Venus^{low} MZB cells have lower CD23 and IgD levels but higher CD1d and IgM levels as well as a higher percentage of Ki-67 expressing cells and early apoptotic cells (Supplementary Fig. 2a-d).

Supplementary Fig. 2b-d: H2B-Venus^{low} cells have a more mature MZB phenotype compared to H2B-Venus^{high} cells. The indicated surface markers (b), percentage of Ki-67+ cells (c) and percentage of viable (7AAD^{neg}Annexin^{neg}), early apoptotic (7AAD^{neg}Annexin^{pos}) and late apoptotic (7AAD^{pos}Annexin^{pos}) cells (d) were determined by flow cytometry.

The marker shift described in Supplementary Fig. 2b and the higher Ki-67 expression (Supplementary Fig. 2c) reflect the development from FoB into MZB cells (Lechner et al., 2021, Gomez Atria et al., 2022; PMIDs: 33597542, 35579963). Therefore, we conclude that H2B-Venus^{low} cells have a more mature MZB cell phenotype and therefore may be “older” than Venus^{high} cells.

To strengthen this finding, we separated the CD21^{high} B cells in three fractions according to their CD23 levels, which are continuously downregulated during the maturation of MZB cells (Lechner et al., 2021; PMID: 33597542). With progressive downregulation of CD23, both the Venus-signal in the H2B-Venus^{high} fraction and the percentage of Venus^{low} cells increased (new Fig. 2c).

Fig. 2c: MFIs of Venus^{high} cells and the percentages of Venus^{low} cells increase in the transition from pre-MZB to CD23⁺ MZB and CD23^{low} MZB cells. The gating of pre-MZB, CD23⁺ and CD23^{low} cells is indicated in the contour plot. The histogram overlay shows the Venus expression in the indicated populations and the gating of Venus^{high} and Venus^{low} cells. The graphs summarize the percentages of the Venus^{low} cells and the MFIs within the gated Venus^{high} peaks in the indicated populations.

Strikingly, both Venus^{high} and Venus^{low} cells express higher levels of Hes1 than FoB cells (new Fig. 2a). The reason could be that in the absence of new Notch signals, H2B-Venus is lost earlier in MZB cells than Hes1 - hence we see changes in Venus levels between Venus-high and -low MZBs, but no obvious changes in Hes1 levels.

Fig. 2a: Venus^{low} MZB cells express slightly higher Hes1 levels than Venus^{high} cells. The graphs compile the MFIs of Hes1 in the indicated populations.

The higher expression of Hes1 in both Venus^{high}-MZB and Venus^{low}-MZB cells in comparison to FoB cells suggests that all MZB cells have a strong Notch2 signal but some display a strongly reduced H2B-Venus stability.

3. Past studies have demonstrated that acute disruption of Notch2 signals leads to loss of MZB (PMID: 24323359). The authors propose that access to Notch2 ligands in the follicle may be important for sustaining the MZ state. If MZB cells lose Notch2 signaling in the anatomic MZ and Notch2 signaling when MZB enter the follicle is important for sustaining the MZ state, why do MZ B cells persist in the absence of CXCR5/Cxcl13 where MZB cannot access the follicle and are present only in the anatomic MZ (PMID: 18037889 and PMID: 15184895).

That is a very important point and we thank the reviewer for bringing it to our attention. Some data argue that induction of Notch signaling mainly occurs inside the follicle, because Dll-1 expressing cells, which are necessary for the generation of MZB cells are present in the follicle and at the follicular side of the marginal sinus but not in the MZ (Fasnacht et al., 2013; PMID: 25311507). However, we have to agree with the reviewer, that the accumulation of MZB cells in the MZ in CXCR5KO mice rather suggests that MZB cells do not need to continuously migrate between the follicle and the MZ to refresh their Notch2 signal. There are different possibilities for the persistence of MZB cells in the MZ in CXCR5KO mice: (i) Different ligands are responsible for the generation and maintenance of MZB cells. Using lacZ insertion mice, it has been shown that Dll-4 is expressed by endothelial cells, including these of the marginal zone sinus, and that Dll-1 is expressed by endothelial cells that are located close to the White Pulp (Tan et al., 1999; PMID: 19217325). Therefore, MZB cells may “refresh” their Notch signal by getting contact to those cells. (ii) Some Dll-1 expressing FDCs, which are normally not located in the MZ migrate into the MZ in CXCR5KO mice (Voigt et al., 2000 Eur J Immunol; PMID: 10671212). (iii) The half-life of MZB cells from CXCR5KO and CXCL13KO mice differs from that of WT MZB cells. Therefore, it cannot be excluded that MZB cells in these mice have a higher turnover and are continuously replaced by newly generated MZB cells as soon as they lose their Notch2 signal. (iv) Not only MZB cells accumulate in the MZ from CXCR5KO and CXCL13KO mice, but also FoB cells that are not dependent on Notch2. The sections of these mice indeed demonstrate that all B cells are located in the MZ (Voigt et al, 2000; Cinamon et al. 2008 and Cinamon et al., 2004; PMIDs: 10671212, 15184895, 18037889), suggesting that the MZ in this case should be at least a mixture of FoB and MZB cells.

Be this as it may, the exact function of Notch-signaling in the maintenance of MZB cells is still an open question, which we cannot answer in the scope of the present manuscript. In the future, we need to learn more about the real half-life of MZB cells, the rate of MZB cell regeneration, as well as the refreshment of the Notch-signal in pre-existing MZB cells.

We are now more careful with our interpretation about the fate of the H2B-Venus^{low} cells. We write in the revised discussion that the fate of H2B-Venus^{low} cells is uncertain and offer 2 alternative possibilities:

(1) the Venus^{low} cells die and are replaced by newly immigrating MZB cells; (2) the Venus^{low} cells migrate into the follicle or to the marginal zone sinus to refresh their Notch signal.

4. How dynamic is Venus expression in Notch2 signaling reporter mice? If ~20% of MZB cells exchange between the follicle and the anatomic MZ over the course of 1 hour, are Venus negative MZ B cells in the anatomic MZ truly an indicator of lack of ligand in this compartment. It would be extremely helpful to understand the dynamics of Notch-reporter expression in B cells in this model in order to know how to interpret the meaning of a reporter negative MZ B cell.

The characterization of H2B-Venus^{low} and H2B-Venus^{high} MZB cells which we performed according to the reviewer’s suggestion revealed that H2B-Venus^{low} cells have a more mature phenotype and slightly higher Hes1 levels than Venus^{high} MZB cells. We are therefore more careful in stating that H2B-Venus^{low} cells are an indicator of lack of Notch ligand in the MZ (further outlined in the answer to point 2 and 3).

There are still several unknown parameters, which limit the exact calculation of the half-life and dynamics of H2B-Venus and its correlation with the intensity of Notch-signaling in MZB cells. We cannot answer the following questions (i) How long does it take until H2B-Venus^{low} cells really lose their Notch-signal (ii)

When do MZB cells receive their Notch maintenance signal: Do they receive it when they are almost H2B-Venus-negative or do they refresh their Notch signal and thus their H2B-Venus-expression when they are H2B-Venus-mid? (iii) Is the refreshment of the Notch signal dependent on the age and maturation stage of the MZB cells? So, it is possible that relatively young MZB cells migrate between the follicle and MZ thereby refreshing their Notch signal, whereas older and more mature cells predominantly remain in the MZ and lose their Notch signal gradually over time. (iii) Do MZB cells start to lose their Venus-signal when they start to differentiate to PC? (iv) What is the fate of Venus^{low} cells? Do they die and are replaced by newly generated MZB cells or do they refresh their Notch-signal further?

We agree with the reviewer that these are very important questions that we plan to address in the future within an upcoming study. Unfortunately, we cannot answer all these questions within the scope of this current manuscript, whose main focus is the generation of MZB cells but not their maintenance.

Our considerations presented above are for the reviewer only, because they are too speculative, to be included in the revised Discussion section.

5. On FACS, Venus expression appears to be uniform in FoB (Venus expression is present in FoB but still at least 5 fold-lower than the highest MZB B cells), how does this uniform amount of reporter expression inform cell fate decisions of FoB transdifferentiating into MZB? A uniform amount of Notch-reporter expression suggests to me that while Notch2 may be required for transdifferentiation it is not sufficient to drive this process and that other signals are more important for guiding cell fate decisions. It would also be helpful for the authors to show MFI values for reporter expression in the various B cell subsets— compared to developing B cells in the bone marrow, reporter expression appears to be ~50-fold higher in 70% of MZ B cells.

how does this uniform amount of reporter expression inform cell fate decisions of FoB transdifferentiating into MZB?

The H2B-Venus expression seems to be rather unchanged in FoB cells. Transitional or FoB cells that finally develop into MZB cells receive an additional strong Notch2 signal, resulting in a higher H2B-Venus expression in MZB cells than in FoB cells. Referring to the reviewer's question we wondered if this upregulation of H2B-Venus is already observed in pre-MZBs, which are characterized as CD23⁺CD21^{high}. Indeed, we observed an intermediate Venus-expression between FoB cells and MZB cells in the CD23⁺CD21⁺ population (**new Fig. 1d**). These cells were originally included in our previously specified FoB cell gate, but since they only account for about 2.5%, they were not visible in the Venus histogram of FoB cells.

Fig. 1d: Venus expression starts to be upregulated in pre-MZB cells. Venus expression was determined in FoB (CD23⁺CD21⁺), pre-MZB (CD23⁺CD21^{high}) and MZB (CD23^{low}CD21^{high}) cells.

A uniform amount of Notch-reporter expression suggests to me that while Notch2 may be required for transdifferentiation it is not sufficient to drive this process and that other signals are more important for guiding cell fate decisions.

Our previous data revealed that switching on constitutive active Notch signaling converts all FoB cells into MZB cells, even in the absence of CD19, suggesting that Notch2 signaling is sufficient for the generation of MZB cells (Hampel et al., 2011, Lechner et al., 2021; PMIDs: 18037889, 33597542). Therefore, we believe that Notch2 signaling is the most important signaling pathway in the generation of MZB cells. In the non-transgenic setting other signaling pathways must cooperate with Notch2, such as PI3K signaling (CD19) and NF- κ B activation. We hypothesize that these signaling pathways are acting upstream of Notch2, resulting in the upregulation of the Notch2 receptor on the cell surface in order to further enhance the intensity of Notch signaling.

It would also be helpful for the authors to show MFI values for reporter expression in the various B cell subsets— compared to developing B cells in the bone marrow, reporter expression appears to be ~50-fold higher in 70% of MZ B cells.

According to the reviewer's suggestion, we inserted all MFIs of developing and mature B cell populations in one graph and show a histogram overlay of pro B, FoB and MZB cells (**new Fig. 1a and b**)

Fig. 1a-b Venus expression is continuously upregulated during B cell development. (a) the graph summarizes the MFI values of H2B-Venus in the indicated populations. (b) The histogram overlay shows the H2B-Venus expression in pro-B, FoB and MZB cells. In comparison to pro-B cells Venus expression is upregulated around 4.7 fold in FoB and 6.6 fold in MZB cells.

6. Although a fraction of GC B cells in the LZ still express Venus reporter, there appears to be no functional consequence to loss of Notch2 signaling on GC dynamics itself. Are the authors suggesting that there is a physiologic function of Notch2 signaling in the LZ driven by FDC or Tfh?

We agree with the reviewer that absence of Notch2 signaling does not have any influence on the germinal center dynamics. Considering the accumulation of H2B-Venus^{high} cells in LZ -B cells, we agree with the reviewer that it would be important to evaluate whether Notch2 signaling has a physiological function in LZ-B cells. Since Thomas and colleagues have shown that stimulation of FoB cells with anti-IgM + anti-CD40 + IL-4 in vitro results in more IgG1-switched cells when the cells are co-cultured on an OP9-DII-1 layer (Thomas et al., 2007; PMID: 17179224), we decided to study the effect of Notch2 deficiency on the expansion of IgG1-switched cells. In fact, we found that the percentage of IgG1 switched GCB cells is

decreased in N2KO//CAR mice in comparison to controls at day 14 p.i.. (**new Fig. 3f and g**). Recently, it has been shown that switching occurs mainly in pre-GC B cells (Roco et al., 2019; PMID: 31375460). Both IgM⁺ and IgG1⁺ B cell blasts differentiate into GCB cells, where they proliferate but do not undergo further CSR. Around 10 days p.i., IgG1 switched cells start to be positively selected at the expense of IgM⁺ cells (Sundling et al., 2021; PMID: 33857421). From this time point onwards more high affinity IgG1⁺ cells than IgM⁺ cells are cycling in the GC. The fact that the percentage of IgG1 switched cells is similar at day 7 after immunization but decreased at day 14 p.i. in N2KO//CAR mice in comparison to controls suggests that Notch2 does not control the switching process to IgG1⁺ cells per se, but rather contributes to the positive selection of IgG1⁺ GCB cells. The Notch2-ligand supporting this process may be provided by FDCs in the LZ-GC, which express the Dll-1 ligand (Santos et al., 2007, Yoon et al., 2009; PMIDs: 17878313, 19542446) (**we discuss this in the discussion of the revised manuscript**).

Fig. 3f-g: N2KO mice have less IgG1⁺ GCB cells. (a) The FACS plots show the gating of IgG1⁺ B cells. The graph below summarizes the cell numbers of reporter⁺IgG1⁺ B cells in control and N2KO mice d7 and d14 post immunization. (b) Reporter⁺IgG1⁺ cells were separated in CD38⁺ (non-GC) and CD38⁻ (GC B cells) cells. The stacked graph summarizes the cell numbers of CD38⁺ and CD38⁻ IgG1⁺ cells in the indicated genotypes 14 days post immunization.

7. In figure 3e (in the new version Figure 4b) the Cg1-cre reporter+CD95+CD38+ cells likely comprise a mixture of both recently activated cells and memory B cells rather than exclusively “activated non-GCB cells” the authors should include additional markers to more fully characterize this cell population or should modify their statement.

As suggested by the reviewer, we performed further stainings to characterize the phenotype of CD38⁺CD95⁺ B cells in more detail. We included the adhesion marker (ICAM-1) and common activation markers (CD80, CD86, ICOS-L), as well as IgD, to differentiate between memory B cells (IgD^{neg}) and activated naïve B cells (IgD⁺). The assumption of the reviewer was right. About half of the cells were IgD^{low} and ICAM^{low} suggesting that they are rather memory cells than activated FoB cells.

Supplementary Fig. 5b: The CD38⁺CD95⁺ population contains a mixture of activated FoB and memory B cells. In the FACS plot, the gates for GC, CD38⁺CD95⁺ and CD38⁺ cells are indicated. The FACS plot is pre-gated on reporter⁺ lymphocytes. The histogram overlay shows the expression of the indicated markers in reporter⁺ GC (red), CD38⁺ (green), CD38⁺CD95⁺ (blue) and in reporter⁻ cells (grey).

We inserted these additional stainings in Supplementary Fig. 5b and modified our statement in the Results section as follows:

Old version: Our in vitro and in vivo data suggested that Notch2 surface expression is induced on activated non-GC cells

Revised version: Notch2 surface expression is induced on CD38⁺CD95⁺ non-GCB cells, containing activated B cells as well as memory B cells.

8. In Figure 4E-G (in the new version Figure 4g+h and Supplementary Figure 6a), the authors conclusions seem to be somewhat of an over-interpretation of the data. The data support the conclusion that when Notch2 signals are not appropriately down-regulated as in the N2IC animals activated cells are arrested in the pre-plasmablast state but it does not necessarily follow that under physiologic circumstances strong Notch2 signaling in FoB promotes a pre-plasmablastic state. In order to support this statement the authors should examine pre-plasmablast/plasma cell responses in the N2KO//CAR mice. If pre-plasmablasts and/or plasma cell responses are not reduced in the N2KO mice then it would appear more likely that the data shown in 4E-G are the result of non-physiologic expression of the N2IC allele. Moreover, mechanistically, why do persistent Notch2 signals in the N2IC mice arrest these cells in a pre-plasmablast state?

In order to support this statement the authors should examine pre-plasmablast/plasma cell responses in the N2KO//CAR mice.

According to the reviewer's suggestion, we immunized N2KO//CAR mice and N2KO//CD19-Cre mice with NP-CGG and analyzed the PC differentiation at different time points after immunization. N2KO mice had less CD138⁺B220^{low} PC than control mice 7 days p.i., whereas the percentages of PC at all other analyzed time points tested were comparable between N2KO and control mice (**new Supplementary Fig. 6b and c**). These data suggest that Notch2 signaling plays a role mainly in the early wave of PC differentiation, when mostly short-lived plasmablasts are generated.

Supplementary Fig. 6b-c: PC are slightly reduced in N2KO mice in comparison to controls. Mice with the indicated genotypes were immunized with NP-CGG and analyzed at the indicated time points p.i.. The FACS plots show an exemplary PC staining 7 days after immunization. The graph summarizes the percentages of PC B220⁻CD138⁺ from different experiments.

A similar result was observed in ELISpot analyses which showed that IgG1-switched ASCs were reduced in N2KO mice in comparison to controls (**new Supplementary Fig. 6d and e**).

Therefore, results in both N2KO settings (γ 1-Cre and CD19-Cre) support the finding from N2IC mice that Notch2 signaling plays a role in driving PC-differentiation. However, our findings from N2KO mice suggest that Notch2 signaling mainly supports the early wave of PC-differentiation.

However, our data revealed only a minor reduction of PC in N2KO mice, which might be due to a redundant function of Notch1 and Notch2 in PC development. Notch1 has been shown to play a role in ASC formation in the presence of LPS and Dll-1 in vitro and in response to immunization with NP-Ficoll and NP-LPS in vivo (Santos et al., 2007; Kang et al., 2014; PMIDs: 17878313, 24913005). **We have inserted this point in the revised discussion.**

Moreover, mechanistically, why do persistent Notch2 signals in the N2IC mice arrest these cells in a pre-plasmablast state?

We show in Figure 5a that mice with sustained Notch2 signaling (N2IC/hCD2) have less Blimp1^{high}IRF4^{high} cells than controls. Upregulation of Blimp-1 beyond an early point is required for terminal PC-differentiation (Shapiro-Shelef 2003; Kallis et al., 2004; Kallis et al. 2007; PMIDs: 9892614, 15492122, 17509907). Moreover, Blimp-1 has been shown to prevent the upregulation of proliferation associated genes (Schaffer et al., 2002; PMID: 12150891) and to act as a tumor suppressor gene (Calado et al., 2010; Mandelbaum et al., 2010; PMID: 21156282, 21156281). We therefore asked whether the plasmablastic B220⁻CD138⁺ cells from N2IC/hCD2 mice proliferate more than the B220⁻CD138⁺ population from control mice. We found that Ki-67 levels were higher in N2IC-expressing plasmablastic cells in comparison to

N2KO and control mice (**new Fig. 5b**). We did not detect a significant difference neither in the Blimp-1 nor Ki-67 expression between N2KO and control mice (new Fig. 5b and **new Supplementary Figure 7a**). So, the effect we see in N2IC/hCD2 mice might be physiologically mediated by Notch1 signaling rather than Notch2 signaling (N2IC is mimicking both Notch1 and Notch2 signaling).

Fig. 5b: Reporter⁺IRF4^{high}B220^{low} cells from N2IC mice proliferate more. The histogram overlay shows the Ki-67 expression in reporter⁻ cells and in reporter⁺ PC (B220⁻CD138⁺) from the indicated genotypes and populations in the spleen. The graph summarizes the MFIs of the Ki-67 expression in PC from control/CAR (open circle), N2KO//CAR (green) and N2IC/hCD2 (blue) mice.

We conclude from these data that Notch signaling prevents terminal PC differentiation and thus keeps the plasmablasts (PB) in a proliferative stage for a longer time. Reduced proliferation of plasmablastic cells could explain the reduced PC numbers during the early wave of PC-differentiation in N2KO mice. Moreover, these data suggested that Notch signaling has to be switched off to allow terminal PC differentiation. We confirmed this hypothesis by showing that H2B-Venus-expression is continuously downregulated in the course of PC differentiation (data **now Fig. 5c and d**). Furthermore, we added a new figure showing that Notch target genes are lower expressed in PB and PC than in FoB and MZB cells (**new Supplementary Fig. 7c**).

Supplementary Fig. 7c: Notch2 and Notch target genes are lower expressed in plasmablasts (PB) and PC in comparison to FoB and MZB cells. In silico analysis of Notch2 and the Notch target genes Dtx, Hes1 and Hes5 in the indicated populations using the immGen database.

9. Figures 5 and 6 offer a nice illustration that both enforced Notch2 signaling can drive MZ differentiation of activated B cells and that the generation of these cells is dependent on post-activation Notch2 signaling. The authors then use mathematical modeling to suggest that these T-dependent MZB cells arise from activated B but not GC B cells. I think the manuscript would benefit from a more in depth analysis of these cells. Can the authors provide additional experimental evidence to define the cellular ontogeny of these cells (I.e. that in WT mice they are not derived from GC B cells)? Are antigen-specific (NP-binding) cells present in T-dependent MZ B cells in similar

numbers to MBC that are not present in the MZ? What is the contribution of these cells to a recall immune response (i.e. in a N2KO mouse is there a reduction in low affinity or high-affinity NP-specific antibody production at memory time point)?

Are antigen-specific (NP-binding) cells present in T-dependent MZ B cells in similar numbers to MBC that are not present in the MZ?

We determined the percentage of NP-specific cells in reporter⁺ MZB cells in comparison to FoB cells and memory B cells. The gating strategy is indicated in the (new Supplementary Fig. 9b)

Supplementary Fig. 9b: Gating strategy for the determination of NP⁺ cells within the indicated subpopulations. CAR⁺ cells were divided in MZB cells (CD21^{high}CD23^{low}) and non-MZB cells. Non-MZB cells were further subdivided in CD38⁺IgD⁺ FoB cells and CD38⁺IgD⁻ memory B cells. The percentage of NP-specific cells was determined in each population and afterwards calculated back to the total percentage of CAR⁺ cells.

The percentages of NP⁺ memory B cells and NP⁺ MZB cells within the fraction of CAR⁺ cells were in a similar range. However, from day 14 to day 26, the percentages of NP⁺ MZB cells significantly increased, while those of FoB and memory B cells remained rather stable in (new Fig. 7c). These data suggest that with ongoing time p.i. reporter⁺NP⁺ cells become enriched in the MZB cell compartment.

Figure 7c: The percentage of NP⁺ MZB cells within the Cre-reporter⁺ fraction increases over time p.i.. First the percentages of MZB and non-MZB cells within the CAR⁺ cells were determined. In the second step the fractions of FoB and memory B cells within the CAR⁺ non-MZBs were gated. After that the NP⁺ fraction within these subpopulations was determined. In the third step the percentages of NP⁺ cells in each population was calculated back to the population of CAR⁺ cells, which was set to 100%.

What is the contribution of these cells to a recall immune response (I.e. in a N2KO mouse is there a reduction in low affinity or high-affinity NP-specific antibody production at memory time point)?

According to the reviewer's suggestion, we re-boosted the mice with NP-CGG/PBS 35 days after the primary immunization with NP-CGG/alum and determined IgM and IgG1 producing NP-specific ASC in the spleen, as well as NP-specific serum titers 7 days after the secondary immunization (Fig. 7d). Total NP-specific IgM and IgG1 serum titers were reduced in N2KO mice in comparison to controls. This may indicate that IgM⁺ MZB cells, which are missing in N2KO mice, actively contribute to the recall immune response in control animals through IgM⁺ and IgG1⁺ PC production. However, it cannot be excluded, that N2KO B cells have a general defect in the memory B cell response independent of the missing MZB cells, which could also explain the reduced titers of NP-specific IgM antibodies and isotype switched NP-antibodies. In the future, further analyses will be necessary to definitively prove that NP-specific MZB cells participate in the recall immune response.

Fig. 7d: NP-specific antibodies in N2KO mice after reboost. (A) The percentages of total (NP14) and high affinity (NP3) NP-specific IgM and IgG1 ASCs in the spleen were determined by ELISpot (B) NP-specific antibodies with the indicated isotype and affinity in the serum were analyzed ELISA.

For the reviewer: NP-specific antibodies in N2KO mice after reboost. (A) The cell counts of total (NP14) and high affinity (NP3) NP-specific IgM and IgG1 ASCs in the spleen were determined by ELISpot

We also performed ELISpot analysis after secondary immunization. The ASCs in N2KO mice were reduced by trend but the differences were not significant.

We included the titers of NP-specific IgG1 and IgM antibodies in the new Fig.7 d. The NP-spec ASCs from the ELISot analysis are information for the reviewer only. Most likely the group of only 4 mice was not big enough to reach a significant result, but there is a strong and rather promising tendency.

Can the authors provide additional experimental evidence to define the cellular ontogeny of these cells (I.e. that in WT mice they are not derived from GC B cells)?

Our future plans are to determine the origin of these cells in more detail, which is currently part of an upcoming study. Thus, we want to analyze the hypermutation rate of CAR⁺ MZB cells at different time points after immunization as well as the clonal relationship between FoB, GC, memory B cells and MZB

cells. Since this important question of the reviewer is directly connected to our follow-up research we believe that it would be better to answer this question in-depth in a follow up paper. To address this point, at least in part, we have mathematically modeled a transplantation experiment that we plan to perform in the future to answer the reviewer's question (new supplementary Fig.12, described in the discussion).

10. It would be helpful for the authors to examine Notch-signaling Venus expression in activated B cells as well as T-dependent MZ B cells. Do they show similar amounts of expression of the Venus reporter as resting FoB and MZB?

Unfortunately, we could not do the proposed experiment, since we did not have CAR// γ 1-Cre mice in combination with the H2B-Venus reporter. The new mating of these mice would have delayed resubmission for many months. Therefore, we decided to analyze the Hes1 protein, as downstream target of Notch-signaling in the different B cell populations instead. Hes1 expression is upregulated in activated B cells and MZB cells in comparison to GC B cells and CD38⁺ cells. We found that the Hes1 expression of MZB cells is in between that of activated FoB cells (reporter⁺CD38⁺CD95^{high}) and reporter⁺CD38⁺CD95^{low} cells and is clearly higher than that of reporter⁻ cells.

Hes1 is higher expressed in reporter⁺ cell populations in comparison to reporter⁻ cells. Within the reporter⁺ cells, Hes1-expression is highest in activated FoB cells (CD38⁺CD95⁺) and MZB cells, while it is lowest in GC B cells.

We have inserted the regulation of Hes1 in CD38⁺CD95⁺, CD38⁺CD95^{low} and GC B cells in Fig. 4C and Supplementary Fig. 5d of the revised manuscript.

The information about Hes1 expression in CAR⁺ MZB cells was not included in the revised manuscript and is only added as additional information for the reviewer.

Minor concerns:

1. MZ B cells in the mouse are non-recirculating but are highly migratory. ~40% of MZ B cells can be visualized exchanging between the follicle and MZ within 1 hour (~20% move between the follicle and MZ and ~20% move between the MZ to the follicle) (PMID: 23263181). Please change the sentence in

the introduction: “FoB cells recirculate between lymphoid organs, whereas MZB cells are mostly sessile in the marginal zone (MZ) and can occasionally migrate between the B cell follicle and the MZ (1)” to more accurately reflect the findings of the citation.

We have changed the sentence according to the reviewer’s suggestion into:

MZB cells are mostly sessile in the spleen, where they migrate extensively between the MZ and the follicle, with approximately 40% of MZB cells exchanging between the MZ and follicle within one hour (Arnon et al., 2013).

2. Data in Figure 3b and c using *cg1-cre/N2IC* mice is largely confirmatory of what has been previously shown by Valls et al (Supplementary Figure 3 in PMID: 28232365). Please indicate this in the text when discussing this data.

We apologize for not properly citing the work of Valls and colleagues. We agree with the reviewer that the result that constitutive active Notch signaling blocks the GC is largely confirmatory. However, our group was describing this phenomenon even earlier. In 2011, we showed that *Notch2IC//CD19-Cre* mice do not form GC upon TD-immunization (Hampel et al., 2011, Supplementary Fig. 7A). Our work was not cited in the manuscript of Valls et al., 2017, which confirmed our results.

To clarify that the result that constitutive active Notch2 signaling blocks the GC reaction is largely confirmatory, we now inserted in the Results section the following sentence:

These findings align with our previous publication and a publication from Valls and colleagues, indicating that constitutive Notch2- signaling is incompatible with the GC reaction (Hampel et al., 2011, Valls et al., 2017).

3. Please label x-axis in all histograms in Supply. Fig. 2b

We apologize for this negligence. We have labeled the x-Axes in the histograms (now supplementary Fig. 4b upon revision).

4. Can the authors examine Notch2 MFI on LZ GCB versus DZ GCB? If CD40/BCR is driving Notch2 expression then it is more likely that Notch2 expression would be increased on LZ. It would also be helpful to see if there is a correlation between Notch2 MFI and Venus reporter expression in LZ and DZ GCB.

We now provide evidence in the revised manuscript that the H2B-Venus-expression and Notch2-surface expression are slightly higher in LZ-B than in DZ-B cells (**new Supplementary Fig. 3c and d**).

Supplementary Fig. 3c-d: The Venus expression and Notch2 expression are slightly higher in LZ compared to DZ B cells. (a) exemplary histogram of the Venus expression in LZ-B and DZ-B cells. The graphs summarize the MFI of Venus (a) and Notch2 (b).

5. The data in figure 4A-C regarding mutual antagonism of Notch2 and Bcl6 has also been demonstrated by Valls et al. Please modify the sentence “These findings indicate that Notch2 signaling interferes with GC formation through Bcl6 suppression” to cite the Valls et al study (PMID: 28232365) appropriately.

We agree with the reviewer that we should be more precise what is new in our paper and what has already been shown in the manuscript from Valls and colleagues.

In the paper of Valls et al., it was clearly demonstrated that Notch2 signaling is downregulated in GC B cells. We took proper consideration of their previous work and therefore largely removed our statements about downregulation of Notch2 signaling in GCB cells from our manuscript. We still show downregulation of H2B-Venus expression in GCB cells, but emphasize that this is consistent with the data from Valls and colleagues, showing that Bcl6 counterregulates Notch2 expression and Notch2 pathway genes in GCB cells. Furthermore, we show the upregulation of Notch2 and Hes1 in activated B cells, and their subsequent downregulation in GCB cells.

However, although it has been very nicely and convincingly shown by Ari Melnick’s group that Bcl6 inhibits the transcription of Notch2 and Notch2 target genes the inverse correlation has not been studied. Thus, to our knowledge, our finding that induction of N2IC in activated B cells inhibits the upregulation of Bcl6 and instead induces Irf4 expression is new. We therefore did not change the sentence but added Irf4-upregulation. We write now: “*These findings indicate that Notch2 signaling interferes with GC formation through Bcl6 suppression and Irf4 upregulation*”. We hope that the reviewer agrees with this.

.

6. When discussing antigen-dependent generation of MZB (page 29), please also cite PMID: 35768001 which has clearly demonstrated MZ-like memory B cells arising following viral infection in mice.

Thank you very much for bringing this article to our attention. We read it with great interest. We mention this study in the discussion (line 413) and have inserted the reference.

Comments to reviewer #3

1) **Are the “ v ” parameters which appear throughout (for example in supplemental equation 1 and main text equations 2 and 6 taken to be the same). If not, can they be assigned separate symbols to avoid confusion.**

Thank you for pointing this out. We have corrected this in the revised version.

2) **It was unclear how the various processes in the model should depend on the existence of Notch2**

signaling. If I correctly understood the details, aside from the decay of the CAR⁺ MZB cells, these parameters seem to be taken to be the same for the “wild-type” and the knockout. It was not clear what the basis was for this assumption. why for example was the μ production rate not dependent on Notch2 signaling given the statement in the discussion that this signaling induces MZB development. In general, the model choices could have been connected better to the underlying biology.

We appreciate the detailed scrutiny of our models, but there seems to be a minor misunderstanding. Indeed, we do consider differences in the generation of CAR⁺ MZB cells between wild-type (control/CAR) and Notch2 knockout (N2KO//CAR) mice. Since, Notch2 is indispensable for MZB cell generation (PMID: 11967543, 12753744, 15146182), we assume absence of differentiation of activated (CAR⁺) B cells – either FoB or GCB – into MZB phenotype (*i.e.*, $\mu = 0$). To emphasize this point, we have modified **Fig. 8 (panels C and D)** and the description of mathematical models in the main text and the methods section. We take great care in developing integrative approaches that closely connect theory and biology, and believe that here too, our model designs are well tethered to the underlying biology of the experimental system and suitable for the questions we are asking.

- 3) **In general, I believe that mathematical models should try to predict new features rather than just recapitulate existing data. Are there any new predictions for further experiments that the authors can propose, as this would increase the usefulness of the modeling section.**

We fully agree with this viewpoint and thank the reviewer for suggesting it. We now include simulations of the immune-response dynamics of MZB cells in an adoptive transfer setting (a direction that we hope to explore in the future) using the rates of their generation and loss estimated from modeling the CAR-expression data. Specifically, we simulated a transplantation experiment in which MZB cell generation from donor FOB and GCB cells (CD45.1 and CD45.2 backgrounds, respectively) is tracked in congenic (CD45.1/CD45.2 background) recipient mice. The virtual transplantation experiment is described in detail in the **new Supplementary Text 3** and schematically in the new **Supplementary Fig. 12**. We simulated the dynamics of donor-derived MZB cells in synchronously immunized recipients, assuming either (i) the branched model, (ii) the linear model, or (iii) a hybrid model in which both branched and linear pathways operate to govern B cell fate decisions. These simulations would help identify critical design components of the adoptive transfer experiment we proposed here, such as the duration of the experiment, selecting intervals between observations, etc. They also provide the means for out-of-sample validation of candidate pathways of B cell differentiation during immune responses by pinpointing conceptual distinctions between their experimental outcomes.

REVIEWER COMMENTS

Reviewer #1 (Remarks to the Author):

The authors responded to most of my criticisms in a more or less convincing fashion. I remain with my major criticisms that using a) NICD2-tg is somewhat artificial and that b) the interpretations with CBF-Venus reporter mice remain not the strongest.

a) The best argument for the authors claim is the expression data of Notch target genes. Otherwise the NICD2-tg expression needs to be taken with a grain of salt.

b) It is very unlikely that under certain conditions histones are lost or degraded in large quantities. It is simply cell division—(or maybe apoptosis/ cell death). Why should a B-cell in normal differentiation degrade the very stable histone octamer? In the chosen LPS-experiment there is cell-divisions AND most likely Notch(2) induction. Thus, this is hard to interpret. Thus, I still very much doubt the 12 hours half-life of H2B-Venus protein stability, let alone that H2B-Venus is less stable than Hes1. The experiment to do is arrest cell-cycle, arrest translation and perform a time-course looking at H2B-Venus. At least the interpretation in the text should be done in more careful manner.

Reviewer #2 (Remarks to the Author):

The authors have carefully responded to my previous critique and the paper is now suitable for Nature Communications.

Reviewer #3 (Remarks to the Author):

I am happy with the responses to my previous report, especially the inclusion of some model-based predictions to be tested in future experiments. I have no further requests

Manuscript *MS # NCOMMS-22-45600*

Reviewer #1 (Remarks to the Author):

We express our gratitude to Reviewer #1 for his/her careful re-evaluation of our manuscript. We have taken great care to address the points he/she raised. We believe that the additional experiment, proposed by Reviewer #1 not only unraveled new understandings of the H2B-Venus stability over time, but also clarified our comprehension of the downregulation of H2B-Venus in some MZB cells. Please find our point-by-point response to the reviewer's points of criticisms below.

I remain with my major criticisms that using a) NICD2-tg is somewhat artificial. The best argument for the authors claim is the expression data of Notch target genes. Otherwise, the NICD2-tg expression needs to be taken with a grain of salt. b) It is very unlikely that under certain conditions histones are lost or degraded in large quantities. It is simply cell division—(or maybe apoptosis/cell death). Why should a B-cell in normal differentiation degrade the very stable histone octamer? In the chosen LPS-experiment there is cell-divisions AND most likely Notch(2) induction. Thus, this is hard to interpret.

Thus, I still very much doubt the 12 hours half-life of H2B-Venus protein stability, let alone that H2B-Venus is less stable than Hes1. The experiment to do is arrest cell-cycle, arrest translation and perform a time-course looking at H2B-Venus.

At least the interpretation in the text should be done in more careful manner.

NICD2-tg is somewhat artificial.

In accordance with the valuable suggestions provided by the reviewers during the first revision phase, we have undertaken substantial measures to enhance the robustness of our study, which we believe improved the quality of our manuscript. We carefully validated the data obtained from the analysis of Notch2IC mice by including analyses performed on Notch2-KO mice and by scrutinizing the regulation of Notch target genes. As a result, our conclusions are no longer solely dependent on the analyses of Notch2IC mice.

Thus, I still very much doubt the 12 hours half-life of H2B-Venus protein stability, let alone that H2B-Venus is less stable than Hes1.

We would like to point out that we did not state that the half-life of H2B-Venus is less than twelve hours in our own experimental system. Instead, in our discussion, we referred to the original publication by Nowotschin et al., 2013 (first description of the Notch-reporter mouse) outlining the H2B-Venus protein disappears in some tissues within half a day. We cited this as an illustrative

example, suggesting that, in certain instances, H2B-Venus appears to have a short half-life. In response to the reviewer's concern, we have modified our discussion concerning the stability of H2B-Venus taken into consideration the findings from the recent experiment we conducted (see below).

The experiment to do is arrest cell-cycle, arrest translation and perform a time-course looking at H2B-Venus.

Editor (Remarks to the Author): We suggest that you perform the control experiment regarding the half-life of H2B-Venus, suggested by R#1.

We thank Reviewer #1 for recommending performing the additional experiment, which clearly showed that in B cells, H2B-Venus is stable for at least 72 hours in non-dividing B cells. Moreover, the experiment showed that in addition to the reduction of H2B-Venus by cell division, PC differentiation leads to a further decrease of the fusion protein.

Experimental design:

As suggested by Reviewer #1, we performed an additional experiment to determine the half-life of H2B-Venus in B cells.

We purified splenic B cells from CBF:H2B-Venus mice and incubated them for 3 days with LPS (i) without inhibitor, (ii) in the presence of 1 μ M Cycloheximide (protein-synthesis inhibitor), or (iii) 1.5 μ M Palbociclib (cell cycle inhibitor G1/S). The optimum inhibitor concentrations were determined in a preliminary experiment. The experiment was not performed in the absence of stimulation (without LPS), because under this condition the ex vivo isolated B cells do not proliferate and almost all die within two days. Sustaining the viability of ex vivo isolated splenic B cells beyond a 3-day period was challenging in the presence of Palbociclib and Cycloheximide even after addition of LPS. The experiment was therefore terminated after 72 hours, and the kinetics of H2B-Venus was followed up to this time point.

To follow cell proliferation, we stained the B cells with CellTrace Far Red. The cells were analyzed after one, two and three days of LPS +/- inhibitor treatment.

Results:

The Venus signal is downregulated by proliferation.

We observed a stable H2B-Venus expression in Cycloheximide treated cells for at least 72 hours. The protein translational inhibitor resulted in complete downregulation of B cell activation (B cells did not increase in size (data not shown)) and cessation of proliferation. This indicates, as suspected by the reviewer, that in B cells H2B-Venus is very stable.

The addition of the cell cycle inhibitor Palbociclib (G1/S phase inhibitor) initially induced an almost complete cell cycle arrest, preserving H2B-Venus stability in the first 48 hours. However, after 2 days, some cells had proliferated and exhibited a decline in their H2B-Venus signal. In the

absence of inhibitors, LPS-stimulated cells experienced a gradual loss of H2B-Venus protein over time due to cell proliferation (Figure 1), supporting the assumption of Reviewer #1 that H2B-Venus is mostly reduced through dilution during cell division. Intriguingly, in both LPS-only and LPS + Palbociclib treated groups, we observed a subset of cells that underwent division, scored by dilution of CellTrace Far Red, but still maintained high levels of H2B-Venus. We speculate that the constant level of H2B-Venus in these cells is driven by the presence of Notch-IC in the nucleus, which leads to the re-synthesis of the protein during proliferation.

In summary, our findings lead us to conclude that in non-proliferating B cells, the half-life of H2B-Venus extends beyond 72 hours. In most B cells, the H2B-Venus protein level decreases depending on the extent of proliferation. However, it is crucial to note that this experiment did not enable the determination of the actual half-life of H2B-Venus in B cells, as it was not feasible to maintain ex vivo isolated B cells beyond 72 hours in the presence of inhibitors or without stimulation.

Figure 1: Venus is downregulated by proliferation. The FACS-plots depict the H2B-Venus expression correlated with the FarRed Cell Trace staining in ex vivo isolated splenic B cells. The cells were subjected to LPS stimulation without an inhibitor, with the cell cycle inhibitor Palbociclib, and the protein synthesis inhibitor Cycloheximide for the indicated time points. The gating strategy involved categorizing the cells into non-proliferating H2B-Venus^{high} cells (non-division + Venus high), cells that remained H2B-Venus^{high} despite proliferation (division + Venus high) and cells that divided while losing their H2B-Venus expression (division + Venus loss). The histogram-overlay at the right side presents the CellTrace Far Red staining under different conditions (red: LPS; blue: LPS + Palbociclib; green: LPS + Cycloheximide). By measuring the dilution of the CellTrace Far Red we determined three cell divisions for LPS-treated

cells after 2 days and 4.5 cell divisions after 3 days. These plots are representative of three independent experiments (= 3 mice). The plots are gated at single living cells.

The Venus signal is downregulated during LPS-induced PC differentiation.

Subsequently, we asked whether in addition to proliferation, H2B-Venus downregulation also occurs during plasma cell (PC) differentiation. We gated cells based on the same CellTrace Far Red intensity - with the intention of examining cells that underwent equal numbers of divisions - and afterwards separated them into three distinct groups according to their CD138 levels: CD138^{low} cells represent PBs dividing cells without differentiation, CD138^{mid} cells were defined as pre-plasmablasts and CD138^{high} cells representing plasmablasts/plasma cells (Figure 2 contour plot on the left side, upper row). In addition, we analyzed within these gated populations the downregulation of B220 (middle row), which occurs throughout PC differentiation. These analyses revealed a correlation between increasing CD138 levels, decreasing B220 levels and increased percentages of H2B-Venus^{low} B cells within cells with the same division rate (Figure 2, single histogram plots (bottom row) and histogram overlay (upper row, right)). These findings suggest that an unknown mechanism contributes to the further downregulation of H2B-Venus during PC differentiation. Consequently, this implies that the observed loss of H2B-Venus in LPS-stimulated B cells is not solely attributed to proliferation but is also influenced by the process of cell differentiation.

Figure 2: Loss of H2B-Venus during PC differentiation. The FACS plot at the upper left presents the gating strategy. Cells were gated based on equal CellTrace Far Red intensity and subsequently subdivided into CD138^{low} cells (B cells div.), CD138^{mid} cells (pre-PB), and CD138^{high} cells (PB). The histograms illustrate the B220 and H2B-Venus-levels in the gated populations. In the histogram overlay displayed in the upper right, the H2B-Venus expression is compared among the indicated gated populations 72 hours after LPS-treatment.

Downregulation of Venus in MZB cells:

The additional experiment suggested by Reviewer #1 let us to suggest an explanation for the differences between the consistently high expression of H2B-Venus in FoB cells compared to its downregulation in some MZB cells. In contrast to FoB cells, MZB cells exhibit self-renewal capacity and exist in a pre-activated state, inherently capable of spontaneous differentiation into plasma cells (PC). Based on the data presented above, we postulate that in some MZB cells H2B-Venus is lost due to their proliferation and differentiation toward PC. Supporting our hypothesis, we present evidence that H2B-Venus^{low} cells have a higher percentage of Ki67⁺ cells than H2B-Venus^{high} cells and FoB cells indicating their stronger proliferation (Figure 3a). In addition, the levels of Irf-4 and Blimp-1 gradually increase from FoB via H2B-Venus^{high} to H2B-Venus^{low} MZB cells, suggesting that H2B-Venus^{low} MZB cells have progressed furthest toward the plasma cell fate (Figure 3b).

Figure 3: H2B-Venus^{low} MZB cells show enhanced proliferation and PC differentiation. Ki-67, Irf4 and Blimp1 stainings were performed with ex vivo isolated splenic B cells. **a)** In the FACS plots the gating strategy for Ki-67⁺ cells is depicted. H2B-Venus^{high} and H2B-Venus^{low} MZB cells were gated as shown in Supplementary Fig. 2a of the revised manuscript. The graph compiles the percentages of Ki-67⁺ cells in the indicated cell populations from different experiments. **b)** The H2B-Venus expression from FoB cells (CD23⁺CD21^{high}) and MZB cells (CD23^{low}CD21^{high} cells) is depicted in the FACS-histograms. The gating of MZB cells and H2B-Venus^{high}, H2B-Venus^{mid} and H2B-Venus^{low} MZB cells is indicated in the FACS-Plot (on the left side) and the H2B-Venus histogram (in the middle), respectively. The graphs on the right side compile the MFIs of IRF4 and Blimp1 in the indicated populations.

In summary, we conclude that overall, the H2B-Venus reporter correlates well with active or recently active Notch signaling. However, we agree with the reviewer that a long persistence of the fluorescent protein must be taken into account, when interpreting H2B-Venus expression in B cells. In contrast to Notch target genes, the Notch reporter serves as a direct indicator of Notch signaling, while Notch-target genes are regulated by various pathways. This divergence may account for the faster loss of H2B-Venus in comparison to Hes-1 in MZB cells. Our new findings in combination with our previous data suggest that H2B-Venus loss occurs in B cells exhibiting robust proliferation in the absence of a Notch signal (e.g., GCB cells) or those undergoing moderate proliferation and differentiation into plasma cells (PC). Suggesting that MZB cells behave similarly as B cells treated with LPS in vitro, we would like to postulate that MZB cells, undergoing division and initiating the PC-differentiation program lose the H2B-Venus expression.

At least the interpretation in the text should be done in more careful manner.

Following the recommendation of Reviewer #1 we are more careful with our interpretation and refrained from asserting a possible degradation of H2B. We have revised our discussion taking into account our new findings and the concerns raised by the reviewer.

We performed the following changes in the manuscript: (indication of pages and lines correspond to the marked version of the manuscript)

1. We inserted an additional Co-author: **Ursula Rambold**. She performed the additional experiment requested by Reviewer# 1.
2. We included parts of the newly performed experiment in the revised version of the manuscript: These data are described in **result section page 6 line 114- 143** and depicted in the **new Supplementary Figure 3a-d**. **a)** shows a kinetics of the loss of H2B-Venus dependent on proliferation in LPS-stimulated B cells as well as the H2B-Venus expression and CellTrace Far Red expression at different time points after LPS-stimulation in histogram overlays; **b)** illustrates the H2B-Venus expression dependent on PC differentiation **c)** shows the percentages of Ki-67⁺ cells in FoB, H2B-Venus^{high} MZB and H2B-Venus^{low} MZB cells (this figure was shifted from the old Supplementary Fig. 2 to the new Supplementary Fig. 3); **d)** shows the Irf4 and Blimp1 in FoB and MZB cells with different H2B-Venus intensities.
3. We removed **Supplementary Fig. 8b (previously 7b)** and the corresponding text in the result section **p.11 line 238-242** because of partial redundancy with the new Supplementary Fig. 3
4. According to the new results and to the comments of Reviewer #1 we changed the **discussion on pages 17-19, lines 379 – 406**.

REVIEWERS' COMMENTS

Reviewer #1 (Remarks to the Author):

The authors have addressed all my criticisms and also revised the text accordingly. The manuscript is now suitable for publication in Nature Communications.